# River temperature response to atmospheric heatwaves is modulated by discharge and meltwater
Amber van Hamel [1,2,3], Joren Janzing[1,2,3] & Manuela Irene Brunner [1,2,3] ✉

Alpine rivers are becoming increasingly exposed to atmospheric heatwaves. Because of their strong relationship with air temperature, rivers can experience persistent heat anomalies, known as riverine heatwaves, which can have serious consequences for river ecosystems and the economy. This study aims to improve our understanding of how river water temperature responds to atmospheric heatwaves by focusing on the interplay of various hydro-climatic variables that can strengthen or weaken the thermal sensitivity of rivers to such events. Our results show that the response of water temperature to atmospheric heatwaves can vary substantially, with only 47% of atmospheric heatwaves leading to riverine heatwaves. Riverine heatwave development can be prevented by positive anomalies in discharge and meltwater, while negative anomalies in discharge strengthen the link between atmospheric and river temperatures. Future changes in these hydro-climatic conditions will likely increase the sensitivity of Alpine rivers to atmospheric heatwaves.

River systems are increasingly exposed to atmospheric heatwaves, which are defined as periods of at least 5 consecutive days during which the temperature exceeds a local, seasonally varying 90th percentile threshold[1]. These atmospheric heatwaves can have a substantial impact on river water temperature because rivers are highly sensitive to changes in air temperature[2–4]. Substantial increases in water temperature in response to atmospheric heatwaves can have serious consequences for river ecosystems[5,6], society[7], and the economy (e.g. energy production[8,9] and fisheries[10]), particularly if water temperature remains extremely high for an extended period of time during so called riverine heatwaves[11]. Examples of the negative impacts of riverine heatwaves include the massive fish die-offs in Alaska, Japan, and the European Alps in 2019, 2021, and 2022, respectively[12–14], as well as the reduction in nuclear power production at the Beznau power plant in Switzerland in the summer of 2025 due to a lack of cooling water[15].

Riverine heatwaves are likely becoming more frequent because of the strong relationship between water and air temperature[11,16–18]. Numerous studies have demonstrated that river water temperature has increased both locally and globally due to global warming[16,19–21]. In addition, the frequency, persistence, and severity of atmospheric heatwaves are increasing[22], which has led to an increase in the frequency, duration, and intensity of aquatic heatwaves including marine[23,24], lake[25,26], and riverine heatwaves[4,11,16,17,27,28].

Even though air temperature is a strong determinant of river water temperature in lowland rivers, it is not the only variable influencing river thermal dynamics[2,3,29]. In addition, river water temperature is modulated by other atmospheric variables (e.g. radiation, humidity), hydrologic conditions (the amount of discharge and the interaction with different water bodies, e.g. lakes and tributaries), processes at the water-streambed interface (e.g. groundwater-surface water interactions), the surrounding environment (e.g. riparian vegetation and shading), and water management (e.g. hydropower production and thermo-peaking, and thermal effluents from industries and urban wastewater)[2,3,30,31]. In small (headwater) rivers in particular, air temperature is not always an accurate indicator of river water temperature, because local factors such as groundwater in- and outflow, soil moisture, snow water equivalent, impoundments, stream size, and shading by riparian vegetation can play an important role too[2,32–35]. Substantial groundwater or meltwater inflow can potentially mitigate the effect of high air temperatures on river warming[36,37], while low discharges during drought can reduce the heat capacity of rivers and make them more prone to warming[2,38]. Furthermore, while air temperature is typically used as an indicator of heat exchange at the air-water interface, other drivers such as radiation and evaporative cooling also become important as air temperature increases[2]. Therefore, a comprehensive understanding of the response of water temperature requires consideration of a wide range of hydroclimatic variables that might influence the link between air and water temperature.

While the direct influence of air temperature on river water temperature has been well documented, the interplay between river water temperature and additional climatic and hydrological factors remains insufficiently explored. This knowledge gap is particularly expressed for

[1]WSL Institute for Snow and Avalanche Research SLF, Davos Dorf, Switzerland. [2]Institute for Atmospheric and Climate Science, ETH Zürich, Zürich, Switzerland. [3]Climate Change, Extremes and Natural Hazards in Alpine Regions Research Center CERC, Davos Dorf, Switzerland. ✉e-mail: manuela.brunner@env.ethz.ch

alpine rivers during atmospheric heatwaves, whose impact on riverine heatwaves could be modulated through this interplay. The response of streamflow to atmospheric heatwaves strongly depends on the amount of snow and glacier ice available for melting[39]. For example, during the summer heatwave of 2003, the streamflow in alpine rivers in Switzerland increased by up to 60–80% as a result of increased glacier melt—potentially leading to cooling—, while non-glaciated rivers mainly showed anomalously low streamflow—potentially intensifying the warming[40]. Piccolroaz et al.[37] investigated the effect of atmospheric heatwaves on river water temperature. They studied the thermal response of 19 Swiss rivers to three strong heatwaves in Central Europe, showing that lowland rivers react sensitively to atmospheric heatwaves. In contrast, high-altitude, snow-fed rivers, as well as regulated rivers receiving cold water from high-altitude hydroelectric reservoirs or diversions, exhibited a dampened thermal response. Similar cooling effects of reservoir regulation (a phenomenon known as thermopeaking) have been documented by Feng et al.[41] for several rivers in Switzerland. While looking beyond air temperature, these studies only considered discharge as an additional variable that could potentially modulate the effect of atmospheric heatwaves on river water temperature. However, they did not consider other hydro-climatic variables, such as meltwater inflow, relative humidity or drought indices. As a result, we still lack a detailed understanding of the main processes that control the link between atmospheric and riverine heatwaves, especially in mountainous regions and across different seasons.

In this study, we test the hypothesis that water temperature is not always following air temperature extremes and that other processes, such as hydrological (e.g. inflow of groundwater and meltwater) and atmospheric conditions (e.g. humidity and radiation), might also play an important role in mitigating or enhancing river warming during atmospheric heatwaves. In doing so, we aim to improve the understanding of how river water temperature responds to atmospheric heatwaves. We quantify the relative importance of a variety of controlling processes going beyond streamflow by including data on a wide range of hydro-climatic processes, such as glacier and snow melt, soil moisture, radiation, humidity, and water availability (i.e. precipitation minus evaporation). Specifically, we explore how these different hydro-climatic conditions strengthen or weaken the relationship

between air temperature and river water temperature, as well as the development of riverine heatwaves. Riverine heatwaves are defined in a similar way as atmospheric heatwaves, namely as periods of at least 5 consecutive days for which the temperature exceeds a local, seasonally varying 90th percentile threshold (see "Methods" section 3.2). Our study area covers 275 catchments in the European Alps, characterized by a large variety of orographic, hydrological, and climatic processes. We standardize all variables to z-scores to enable a comparison between catchments in space and across seasons (see "Methods" section 3.3). This allows us to compare the severity and extremeness of the observed water temperature with, for example, that of the atmospheric heatwave. Additionally, we analyse two case studies in which we adopt a network perspective to demonstrate how riverine heatwaves propagate downstream along river networks from high-elevation glaciated regions to lower-elevation rainfall-dominated regions under the influence of large lakes (see "Methods" section 3.4).

Our results show that the response of water temperature to atmospheric heatwaves can vary substantially across events and catchments: only 47% of the atmospheric heatwaves observed in the study domain between 2011 and 2021 led to riverine heatwaves. The development of riverine heatwaves can be mitigated by positive anomalies in discharge and meltwater, while negative anomalies in discharge enhance the link between atmospheric and river temperature. These results suggest that future changes in hydro-climatic conditions, including a decline in meltwater and discharge in summer, will likely increase the sensitivity of rivers to atmospheric heatwaves.

## Results and discussion
### Atmospheric heatwaves do not necessarily lead to riverine heatwaves

Atmospheric heatwaves in the European Alps during the period 2011–2021 were most prevalent during the summer months and in low-elevation catchments (Fig. 1a), despite the fact that they could occur equally often throughout the year because of their definition, which relies on a seasonally varying threshold. The only exception from this general pattern is catchments with a mean elevation above 2000 m, which experience slightly more atmospheric heatwaves in winter than in summer (Fig. 1a).

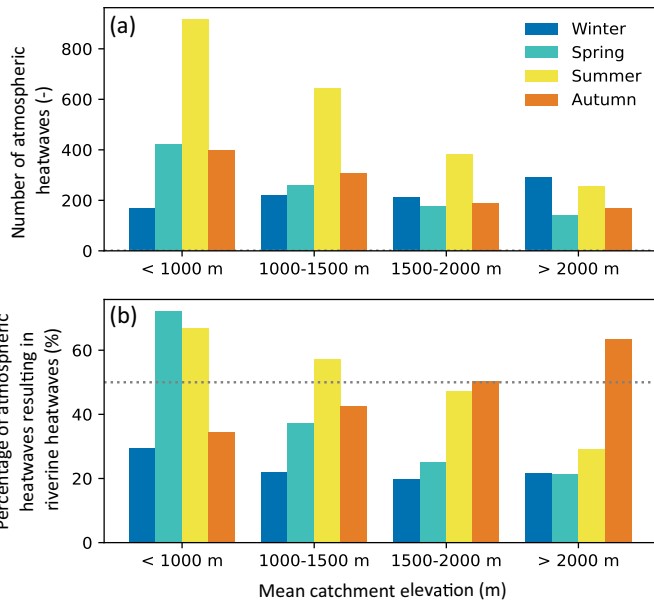

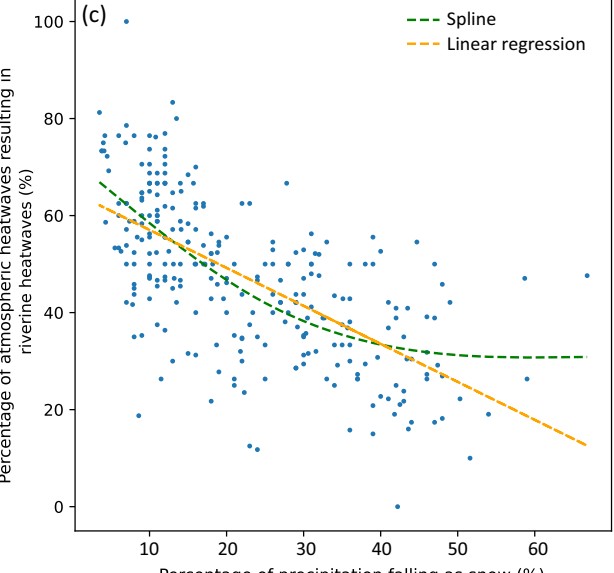

**Fig. 1 | Seasonal and elevation-dependent co-occurrence of atmospheric and riverine heatwaves. a** Number of extracted atmospheric heatwaves and **b** the percentage of these atmospheric heatwaves resulting in a riverine heatwave, grouped per elevation band and per season. The dashed horizontal line highlights a probability of 50%. The number of atmospheric heatwaves per elevation band is: <1000 m, $n = 1901$; 1000–1500 m, $n = 1430$; 1500–2000 m, $n = 959$; >2000 m, $n = 855$.

**c** Relationship between the percentage of precipitation falling as snow per catchment and the percentage of atmospheric heatwaves resulting in riverine heatwaves (Spearman correlation coefficient = –0,668). The linear regression (orange) and smoothing spline with 2 degrees of freedom (green) both highlight the significant relationship between the two variables ($p$ value < 0.05), whereby the spline slightly outperforms linear regression.

Our results show that approximately half of these atmospheric heatwaves (47%) coincide, at least partially, with riverine heatwaves (Fig. 1b). Such coincidence results from a strong coupling between air and water temperatures as indicated by similarly high five-day mean air temperature (AT) and water temperature (water temperature) z-scores. The link between air and water temperatures is strongest in catchments at low elevations with a small snow fraction (Fig. 1c). In these catchments, the probability that an atmospheric heatwave in spring or summer—the season with the highest probability of occurrence of atmospheric heatwaves (Fig. 1a)—will lead to a riverine heatwave is greater than 60% (Fig. 1b and c). At higher elevations ( >2000 m), the likelihood of riverine heatwaves occurring as a result of atmospheric heatwaves is with 60% highest in autumn (Fig. 1b). Although the absolute number of atmospheric heatwaves observed in autumn in high elevation catchments is relatively small (Fig. 1a), these heatwaves appear to have the greatest influence on river water temperature. The probability of atmospheric heatwaves leading to riverine heatwaves in high elevation catchments is relatively low during the rest of the year (probabilities 20–30%). When a riverine heatwave occurs in close temporal proximity to, or simultaneously with, an atmospheric heatwave, it tends to start on the same day or within one to two days of the atmospheric heatwave (Supplementary Fig. 1). In 61% of the cases, the riverine heatwave lasted longer than the atmospheric heatwave, with an extended duration of 1-5 days on average. For more details on the total number of identified atmospheric and riverine heatwaves and their duration, we refer the reader to the Supplementary Information (Supplementary Section 1 and Supplementary Fig. 1).

Regardless of the development of a riverine heatwave, atmospheric heatwaves can result in both positive and negative anomalies in water temperature. During atmospheric heatwaves, which are characterized by a five-day mean air temperature z-score of at least 1.29 (corresponding to the 90th percentile), the corresponding water temperature z-score can vary widely (Fig. 2a). The water temperature z-score can reach similarly extreme values as the air temperature z-score (up to 3.48), but average and negative water temperature z-scores are also possible (down to –1.56) (Fig. 2a). Similar water temperature and air temperature z-scores suggest a strong coupling between air and water temperature, whereby water temperature experiences similarly high positive anomalies as air temperature. This could result in the development of a riverine heatwave. Conversely, a significant difference between the air and water temperature z-scores indicates a reduction in the thermal sensitivity of water to air temperature because the water temperature is barely affected by the high air temperature anomalies. A wide range of water temperature z-scores corresponding to high air temperature z-scores can be observed, including negative water temperature z-scores (Fig. 2b), regardless of catchment elevation or season. However, a slightly larger spread, and thus more negative water temperature z-scores, is found in spring and, for high-elevation rivers ( >2000 m), in summer.

## Hydro-climatic conditions modulate the link between water temperature and atmospheric heatwaves

Atmospheric heatwaves that lead to (or partly overlap with) riverine heatwaves are characterized by different hydro-climatic conditions compared to those associated with atmospheric heatwaves that do not lead to riverine heatwaves (Fig. 3). In general, all atmospheric heatwaves co-occur with normal to slightly negative z-scores in relative humidity, water availability (i.e. precipitation minus evaporation), soil moisture, meltwater fraction, and discharge, while they co-occur with mainly positive z-scores of surface net solar radiation (Supplementary Fig. 2). The atmospheric heatwaves that lead to riverine heatwaves (dark brown events, Fig. 3) are related to significantly lower z-scores in discharge and meltwater fraction, slightly lower z-scores in soil moisture and radiation, and slightly higher (less negative) z-scores in relative humidity and water availability as compared to the atmospheric heatwaves that do not lead to riverine heatwaves (light brown events). A Mann–Whitney U test has shown that there is a significant difference (p < 0.05) between the two event sub-samples (atmospheric heatwaves with and without riverine heatwaves) for all variables. The differences between the two groups is clearly larger for discharge and the meltwater fraction than for

the other variables. This suggests that these other variables are less decisive for the development of riverine heatwaves from atmospheric heatwaves than discharge and melt water contributions.

Discharge is one of the most important variables in determining whether an atmospheric heatwave translates to a riverine heatwave. While discharge anomalies can be positive or negative during atmospheric heatwaves, they are mostly negative when atmospheric heatwaves lead to riverine heatwaves (Fig. 3). Consequently, anomalously high discharge during an atmospheric heatwave can prevent the development of riverine heatwaves: atmospheric heatwaves that do not lead to riverine heatwaves show higher and mostly positive discharge z-scores. This effect of positive discharge anomalies on mitigating riverine heatwaves is most evident in spring and summer (Fig. 4a) and in catchments with a mean elevation between 1000 and 2000 m (Fig. 4b). In autumn and winter, the opposite effect is visible, with higher positive discharge anomalies favouring the development of riverine heatwaves.

The mitigating effect of positive discharge anomalies on water temperatures in spring and summer seems linked to the meltwater fraction that can potentially reach the river, which refers to the amount of meltwater in relation to the sum of meltwater, precipitation, and evaporation. In catchments with snow and ice, substantial meltwater inputs to the river can occur during the melting season in spring and, at higher elevations, also in summer (Fig. 4c, d). Though not always, positive anomalies in the meltwater fraction can result in a significant reduction of the water temperature z-scores and can prevent a river from experiencing a riverine heatwave, especially in combination with positive discharge anomalies (Fig. 4e).

The other hydro-climatic variables are more difficult to interpret in relation to their mitigating or enhancing effect on the development of riverine heatwaves during atmospheric heatwaves. Other potential drought indicators, such as water availability (precipitation minus evaporation) and soil moisture, show a rather mixed behaviour across seasons and at different elevation bands (Supplementary Fig. 4e, f, m, n). A clear, drought-related effect similar to that observed for discharge cannot be found.

Finally, the development of riverine heatwaves seems to be favoured by atmospheric conditions characterized by slightly lower anomalies in surface net solar radiation but higher and more positive anomalies in relative humidity (Supplementary Fig. 4c, d, k, l). High anomalies in relative humidity play a particularly important role in autumn and winter, as well as at higher elevations. In other words, riverine heatwaves occur more frequently in these seasons when atmospheric heatwaves are accompanied by relatively low energy input (radiation) but high humidity (and potentially rainy weather).

## Water temperature response to atmospheric heatwaves changes along a river network

To understand how the water temperature response to atmospheric heatwaves can vary along river networks, we looked at two case studies. In the first case study, we compared the water temperature response of five different stations along the Drau river (Austria) to two similarly severe atmospheric heatwaves in June 2015 and 2017 (see Methods section 3.4). The 2015 heatwave occurred between 3rd and 7th of June and affected all five stations along the Drau river network, with similarly high air temperature z-scores recorded at all stations (Fig. 5a and e, in purple). During this heatwave, the entire river showed a substantial warming, as highlighted by the simultaneous increase in water temperature z-scores across all stations during this period (Fig. 5f). However, the river's water temperature did not reach z-scores similar to those of air temperature, with the result that no riverine heatwaves were observed at any of the stations (Fig. 5b, in purple). Two years later in 2017, another atmospheric heatwave was observed between the 19th and 24th of June. This heatwave affected all stations, but was only labelled as such at the three lowest stations (Fig. 5a and i, in pink). The air temperature z-scores observed during this heatwave were very similar to those observed during the June 2015 heatwave. However, unlike the 2015 event, this atmospheric heatwave resulted in riverine heatwaves at all five stations (Fig. 5b, pink squares).

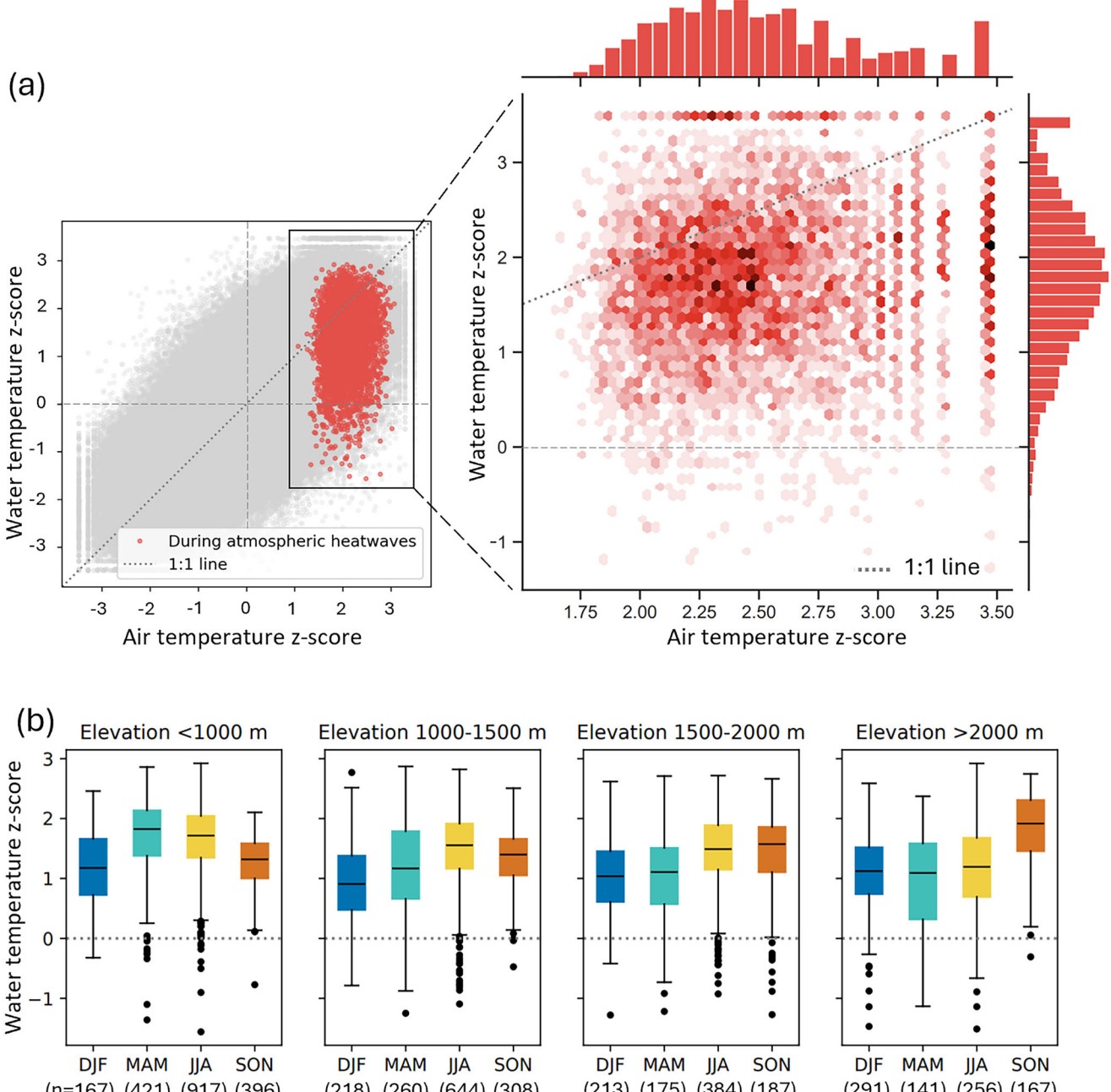

**Fig. 2 | Seasonal response of water temperature z-scores during atmospheric heatwaves. a** Water temperature and air temperature five-day mean z-scores for all time steps and catchments (in grey) and the maximum five-day mean z-scores during atmospheric heatwaves (in red), with a zoom-in density plot on the right. **b** Spread in average five-day mean water temperature z-scores during atmospheric heatwaves, categorised by catchment mean elevation and season: December-February (DJF), March-May (MAM), June-August (JJA), September-November (SON). The boxes represent the interquartile range, with the line indicating the median. The whiskers extend to points up to 1.5 times the box range. Outliers are plotted as individual points beyond the whiskers.

To understand why the response of water temperature to these relatively similar atmospheric heatwaves is so different, we need to consider the discharge and meltwater during these events. In June 2015, positive anomalies in discharge were observed at all stations during the atmospheric heatwave (see Fig. 5c, g), whereas in June 2017, discharge showed much lower positive anomalies or even negative anomalies at the most downstream station (Fig. 5c, k). A similar pattern is found for meltwater (Fig. 5d, h, l): positive anomalies were observed at all stations during the 2015 event, whereas mostly negative anomalies were observed during the 2017 event. In other words, the event in 2015 was characterized by anomalously high meltwater and discharge, while the 2017

event was characterized by low discharge and meltwater because of an already disappearing snowpack. The effect of these differences in discharge and meltwater on water temperature is striking: in 2015, no riverine heatwave developed at any station, while in 2017, riverine heatwaves developed at all stations, including those that did not even experience an atmospheric heatwave.

In the second case study, we assess the influence of lakes on the water temperature response. We compare the water temperature z-scores directly upstream and downstream of five big lakes (Fig. 6) during the seven year-round most severe atmospheric heatwaves observed in our study domain, i.e. those that affected more than 50% of the catchments in our study

**Fig. 3 | Z-scores of different hydro-climatic variables during atmospheric heatwaves.** Comparison of the z-scores for atmospheric heatwaves that lead to riverine heatwaves (dark brown boxplots, $n = 2382$) and those that do not (light brown boxplots, $n = 2763$). All pairs showed significant statistical difference based on the Mann-Whitney U test ($p < 0.05$). The boxes represent the interquartile range, with the line indicating the median. The whiskers extend to points up to 1.5 times the box range. Outliers are plotted as individual points beyond the whiskers.

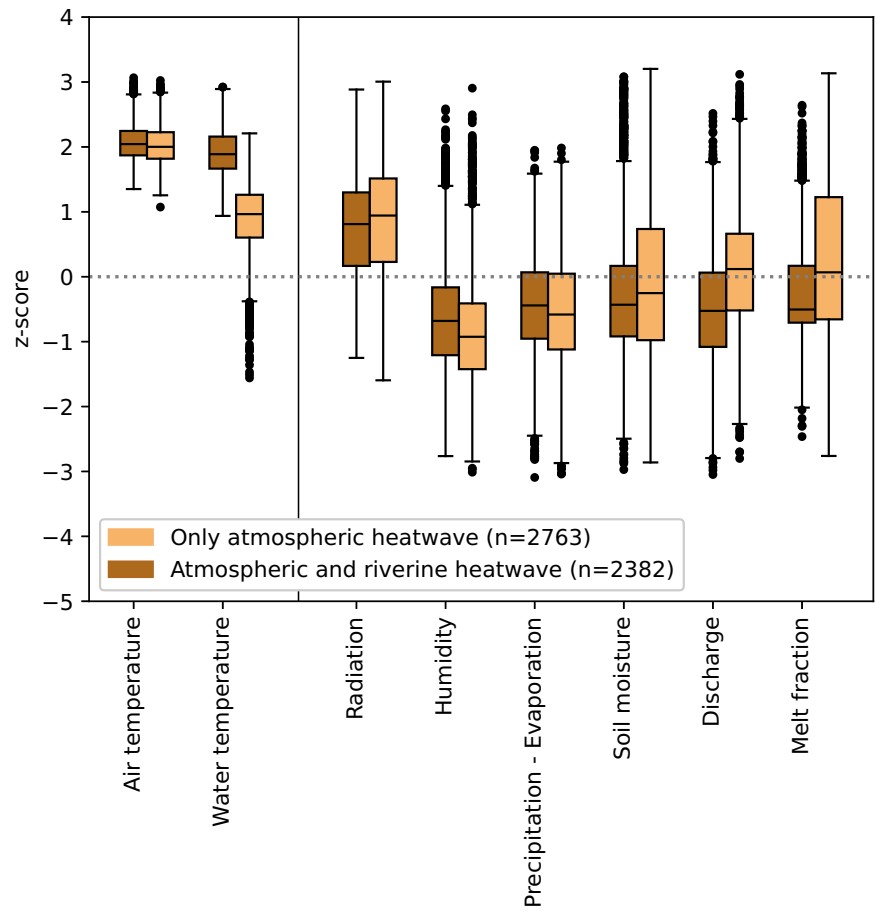

domain. The observed lake effect on the downstream river temperature depends on the season: in winter and spring (see dots), the water temperature z-scores downstream of the lakes are in general lower than those upstream of the lakes. In other words, the rivers downstream of the lakes have warmed less (compared to average water temperature) than those upstream of the lakes. In mid-summer (triangles) and late-summer (crosses), the water temperature z-scores tend to show an opposite sign with higher water temperature z-scores downstream of the lakes than upstream of the lakes. In this season, the warming of the rivers is stronger downstream of the lake than upstream.

**Water temperature sensitivity to atmospheric heatwaves**
Our results show that about half (47%) of the atmospheric heatwaves observed in the European Alps over the period 2011–2021 led to riverine heatwaves, indicating a strong coupling between air and water temperatures (Fig. 1b). This coupling is elevation and season-dependent: it is strongest in spring and summer at low elevations, while it is strongest in autumn at high elevations (Fig. 2b). A similar elevation-dependent coupling has been observed by Piccolroaz et al.[37], who documented high air-to-water temperature sensitivity in low-land rivers, but a complete disconnection between the two variables for high-elevation snow-fed and regulated rivers using 19 catchments in Switzerland.

While half of the atmospheric heatwaves led to riverine heatwaves, the other half did not (Fig. 1b). For these events, the thermal sensitivity of water to air temperature is weak, preventing the development of riverine heatwaves. Our results show that positive anomalies in discharge and meltwater have a clear cooling effect on rivers (Figs. 3, 4e), thereby corroborating similar findings of previous studies[11,37,41]. Such positive discharge anomalies increase the thermal capacity of a river (the ability to absorb heat), which results in a weakened response of water temperature to atmospheric heatwaves. Several studies[2,11,19,42] have shown a similar

negative correlation between absolute discharge and water temperature. While these studies have highlighted the effect of discharge on water temperature from an absolute perspective, our results show that this buffering effect also exists when looking at seasonal discharge anomalies (Figs. 3, 4a, b). This suggests that even under low absolute seasonal flow, positive deviations from the average seasonal flow can substantially impact the thermal sensitivity of river water to air temperature extremes and mitigate the development of riverine heatwaves.

Positive discharge anomalies are typically associated with a seasonal precipitation surplus or melt events. During spring and summer heatwaves, characterised by high pressure weather systems and dry conditions[43–45], meltwater is likely the main cause of positive streamflow anomalies[40]. The relatively cold meltwater input can mitigate the development of riverine heatwaves by lowering water temperatures. Figure 4e highlighted the clear link between positive discharge and meltwater anomalies, whereby a combination of the two can lead to substantially reduced water temperatures. Although the impact of meltwater on water temperature has previously been suggested by others (e.g.[34,36,37,41]), we were here able to provide large-scale, data-based evidence for this effect.

In autumn and winter, an opposite signal is visible, with positive discharge anomalies favouring the development of riverine heatwaves (Fig. 4a). At this time of the year, positive discharge anomalies are less likely to be related to meltwater, which is mostly available in spring, and instead indicate wet conditions, with rivers being fed by warmer rainfall or groundwater. Our finding that riverine heatwaves in these seasons occur more frequently when atmospheric heatwaves are accompanied by lower radiation but higher humidity anomalies (Supplementary Fig. 4c, d, k, l) also supports the hypothesis that rainfall plays a role in the development of positive discharge anomalies. Lower radiation and higher humidity could point towards cloudy and rainy weather, with warm air advection from the south-west entering the region under the influence of low pressure systems[46,47]. This

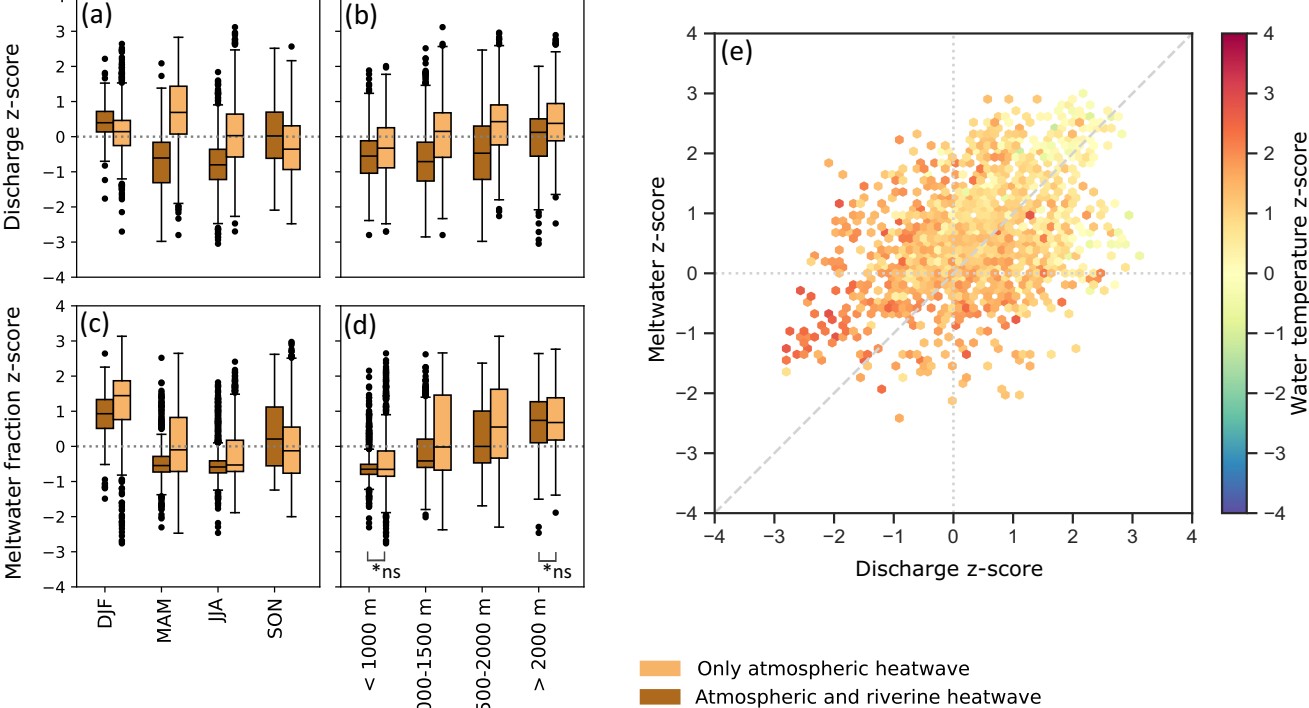

**Fig. 4 | Z-scores of discharge and meltwater fraction by season and elevation band.** Panels **a** and **b** show the z-scores for discharge by season (winter (DJF) with dark brown $n = 202$ / light brown $n = 687$), spring (MAM) with $n = 475/522$, summer (JJA) with $n = 1238/963$, and autumn (SON) with $n = 467/591$) and by elevation band (grouped by mean catchment elevation: <1000 m with $n = 1102/799$, 1000–1500 m with $n = 645/785$, 1500–2000 m with $n = 361/595$, > 2000 m with $n = 274/581$). Panels **c** and **d** show the z-scores for the meltwater fraction by season and elevation band. The light and dark brown boxplots represent atmospheric heatwaves that did not and did result in riverine heatwaves, respectively. The boxes represent the interquartile range, with the line indicating the median. The whiskers extend to points up to 1.5 times the box range. Outliers are plotted as individual points beyond the whiskers. Two pairs that are marked with *ns did not show significant statistical difference based on the Mann–Whitney U test ($p > 0.05$). **e** Discharge z-scores (x-axis) against meltwater z-scores (y-axis) presented by hexagonal bins that represent the mean water temperature z-scores of the data points within each hexagonal bin. See Supplementary Fig. 3 for the density plot.

rainfall can result in groundwater recharge and activate groundwater flow to the river. The inflow can provide the river with relatively warm water in winter[48,49], because groundwater temperature is known to be more stable throughout the year. This could lead to anomalously high river water temperatures and the development of riverine heatwaves.

In contrast to positive streamflow anomalies in summer, negative anomalies, indicative of drought conditions, can support the development of riverine heatwaves (Fig. 3a). This corroborates findings by Tassone et al.[11] who found that riverine heatwaves are often associated with normal or below-normal discharge conditions. Extreme low flow conditions can increase the river water temperature sensitivity to heat fluxes by reducing the river's thermal buffering capacity (e.g. reducing flow depths and flow velocities but enhancing residence times)[38]. The resulting increase in river water temperature can cause a deterioration in water quality, with lower dissolved oxygen levels, increased algae growth, and higher concentrations of pollutants from point sources[50–52].

Low flow anomalies in summer and spring are often associated with dry atmospheric periods, which are related to high atmospheric energy inputs, little water availability with precipitation deficits and excessive evaporation, and low groundwater levels[38,53]. This reduction in groundwater levels and stream-groundwater connectivity can make rivers more sensitive to atmospheric warming[48,49] and consequently favour the development of riverine heatwaves. While a lack of groundwater contributions to streamflow may favour the development of riverine heatwaves, we did not explicitly consider groundwater influences here because of a lack of adequate observed groundwater data. In order to explicitly consider such influences, a proxy for groundwater could be derived from streamflow (e.g. based on a baseflow analysis) or using a hydrological model.

The findings from our two case studies at a local scale support the insights obtained from our large-scale study: high meltwater and discharge anomalies can dampen the warming effect of rivers during atmospheric heatwaves, thereby mitigating the development of riverine heatwaves (Fig. 5). In addition, these case studies highlight the impact of local processes, like the interaction with large lakes, on river thermal sensitivity (Fig. 6). Seasonal differences in the thermal stability of lakes (i.e. the strength of thermal stratification, as discussed in North et al.[54]) have been shown to strongly influence the heat distribution of lakes and the warming of downstream rivers: low thermal stability of lakes during the winter and early spring results in vertical mixing and a better distribution of heat, while the high thermal stability of lakes in summer causes stratification, with a thin upper layer of water that is strongly influenced by atmospheric conditions (and thus stronger warming of the downstream river).

Finally, 69% of the riverine heatwaves in our study could not be directly linked to an atmospheric heatwave. This finding is consistent with findings of van Hamel and Brunner[32], who demonstrated that extreme water temperatures often (in 59% of the cases) result from a combination of non-extreme hydro-climatic conditions rather than from extreme air temperatures alone. Tassone et al.[11] have similarly shown that water temperature trends in catchments in the United States frequently (in 39% of the cases) exceeded the magnitude of atmospheric temperature trends, which also highlights that water temperature is not governed by atmospheric temperature alone. These findings raise the question of what causes those riverine heatwaves if it is not positive air temperature anomalies. Although we identified two important processes that can greatly impact the relationship between water and air temperatures, namely discharge and snowmelt, further research is required to understand the influence of additional

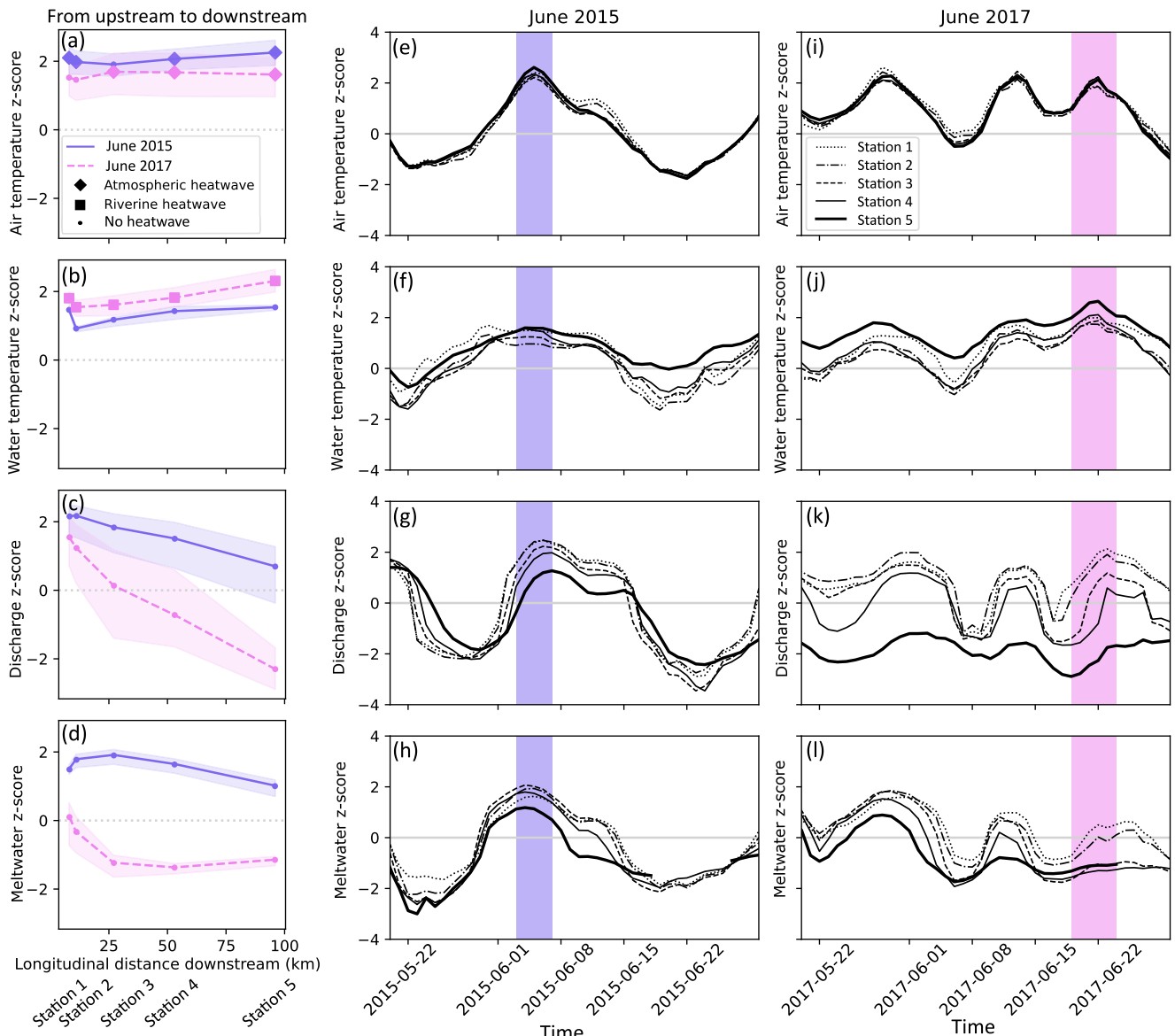

**Fig. 5 | Downstream tracking of the hydro-climatic response to two atmospheric heatwaves along five stations in the Drau river. a–d** The maximum z-scores per station (from the upstream station 1 to the most downstream station 5) are shown for the two events: June 2015 in purple and June 2017 in pink. When stations experience an atmospheric or riverine heatwave, they are labelled with a diamond or square, respectively. Temporal variations in air temperature, water temperature, discharge and meltwater z-scores for five stations along the Drau River during the atmospheric heatwave of June 2015 (**e–h**), and June 2017 (**i–l**).

processes on the development of extreme water temperatures and riverine heatwaves.

**Limitations and future directions**
Together, the large-sample analysis and the two case studies allowed us to gain insights into the factors influencing riverine heatwave development based on observational data. While observations are considered the most reliable data source available, moving to the modelling world could complement the observation-based analyses performed in this study. Numerical water temperature models enable targeted modelling experiments and sensitivity tests to isolate and quantify the influence of specific atmospheric and land-surface conditions on riverine heatwave development. However, a clear isolation of individual physical effects also remains challenging in the modelling world because of different sources of uncertainty involved in the modelling process. Specifically, disentangling the influence of in-stream processes (e.g. streambed friction, turbulence, and river heat capacity) and local conditions around the stream remains challenging and it is yet unclear

which of the two has a larger influence. Furthermore, it remains challenging to account for human influences, such as the effect of reservoir regulation, water withdrawals, and thermal effluents from industries, on the relationship between air and water temperatures because of a lack of consistent information on regulation, particularly across the large, cross-border sample of catchments considered in our study. Such influences could in future studies be investigated using targeted modelling experiments.

Our findings demonstrate that a simplistic air-to-water temperature relationship is insufficient to capture river water temperature dynamics under extreme conditions. Therefore, we argue that, in order to model or predict water temperature along rivers, information beyond the two variables of absolute air temperature and discharge is required. Still, commonly used statistical water temperature models - which are an attractive, less computationally demanding alternative to process-based models - often rely solely on these two types of inputs (e.g.[55–57]). Understanding the spatial and temporal dynamics of water temperature and the development of riverine heatwaves requires novel scientific approaches that recognise the complex

**Fig. 6 | Lake influence on water temperature during atmospheric heatwaves.** Water temperature z-score upstream and downstream of five big lakes in Switzerland during the seven most severe atmospheric heatwaves. The dashed line shows the 1:1 line.

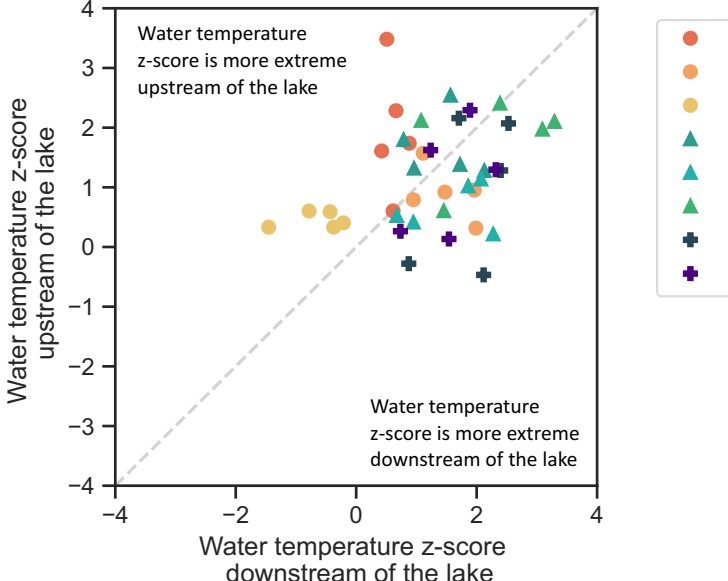

interactions of all mechanisms involved, potentially calling for the use of machine learning or hybrid modelling approaches[58,59].

**Implications for water temperature response in a warming world**
Our main findings highlight the strong link of riverine heatwave development with atmospheric heatwave occurrence, streamflow drought conditions, and meltwater availability. These findings suggest that the frequency of riverine heatwaves will likely increase in the future because the number of potential triggering events (i.e. atmospheric heatwaves and streamflow droughts) is projected to increase[22,60,61], while the frequency of hydro-climatic conditions that mitigate the development of riverine heatwaves (i.e. meltwater peaks) is projected to decrease[62]. A plethora of studies has shown that atmospheric heatwaves will become more frequent, intense, and persistent in the future[22,63–67]. The same applies to the simultaneous occurrence of prolonged streamflow droughts and atmospheric heatwaves[45,68,69].

Furthermore, global warming will result in a shortening and shifting of the melt season[36,70]. Since meltwater has a cooling effect on river water temperature, a shift in the meltwater regime will result in less (snow and glacier) meltwater during late spring and summer. In addition, the discharge regime will shift in some regions from nival (dominated by snowmelt) to pluvial (dominated by rainfall), resulting in even lower summer discharge[62]. All of these projected changes may result in more frequent riverine heatwaves, especially in summer and at low elevation, where the probability (60%) of atmospheric heatwaves translating to riverine heatwaves is highest (Fig. 1b). At the same time, we also expect an increase in the frequency of both atmospheric and riverine heatwaves in winter, as Beniston[46] has shown that winter warm spells in Switzerland may increase by 30%. In conclusion, the combined reduction in meltwater and discharge will likely increase the sensitivity of Alpine rivers to atmospheric heatwaves, with associated negative consequences for energy production, society, and river ecosystems.

## Methods
### Study area and data
This study includes 275 catchments in Switzerland and Austria (Fig. 7a) for which hourly water temperature and discharge data were available for the period 2011–2021. These catchments cover different elevations and sizes, with mean catchment elevations varying from 191 to 2929 m.a.s.l. and catchment areas varying between 10 and 36404 km². For the 207 Austrian catchments, hourly water temperature and discharge observations per station were provided by the Department of Water Management of the Federal Ministry of Agriculture, Forestry, Regions and Water Management. Catchment characteristics were obtained from the LArge-SaMple DAta for

Hydrology and Environmental Sciences for Central Europe dataset (LamaH-CE)[71]. For the 68 Swiss catchments, hourly water temperature and discharge station observations were provided by the Federal Office of the Environment (FOEN), and the catchment characteristics were obtained from CAMELS-CH[72]. To better compare discharge among the different catchments, we used area-independent discharge (mm d⁻¹).

In addition, we obtained data on hydro-climatic variables from the gridded Copernicus European Regional ReAnalysis (CERRA) dataset[73], which has a high temporal and spatial resolution of 3 h and 5.5 km, respectively, for the period 2011–2021. CERRA has been shown to outperform lower-resolution reanalysis products such as ERA5 and ERA5-Land[74]. In addition, it provides homogeneous data across country borders. From CERRA, we extracted catchment mean daily air temperature (˚C), surface net solar radiation (J m⁻²), relative humidity (%), actual evaporation (mm d⁻¹), precipitation (mm d⁻¹), and liquid volumetric soil moisture (m³ m⁻³). Drought conditions are described by proxies such as water availability (i.e. precipitation minus evaporation) and soil moisture, in addition to discharge observations. Gridded daily snowmelt (mm d⁻¹) and glacier melt (mm d⁻¹) were simulated at a 30 arcsec (approx. 1 km at the equator) resolution with the gridded global hydrological model PCR-GLOBWB 2.0[75,76] for the period 2011–2021 and together they present the total daily melt. Specifically, we used an updated version of this model that has been set up and evaluated for the Alps by Janzing et al.[77] and contains an updated snowmelt routine (with an expanded temperature index model and regional calibration) and a recently developed glacier routine. To ensure spatial consistency in snow data across country borders, we used a large-scale snow product generated using a model with a temperature-index based snow routine that has been thoroughly evaluated over the Alpine domain[77], instead of country-specific products that rely on more detailed process representation. As input, the PCR-GLOBWB 2.0 model uses gridded precipitation and downscaled air temperature data. To obtain the downscaled air temperature product from CERRA, we followed the downscaling approach of Van Jaarsveld et al.[78]. We calculated correction factors by comparing monthly climatologies (1990-2019) for the CERRA temperature with the climatologies of the high-resolution Climatologies at High resolution for the Earth's Land Surface Areas (CHELSA) V2.1 data set[79,80]. In our study, we use the catchment mean daily meltwater (mm d⁻¹), defined as the sum of snowmelt and glacier melt, and the meltwater fraction (-), defined as meltwater / (meltwater + precipitation - evaporation). We do not explicitly account for catchment-specific time lags between temperature-induced melting and runoff routing, since these lags are assumed to be very short (shorter than one day for most headwater catchments and within five days

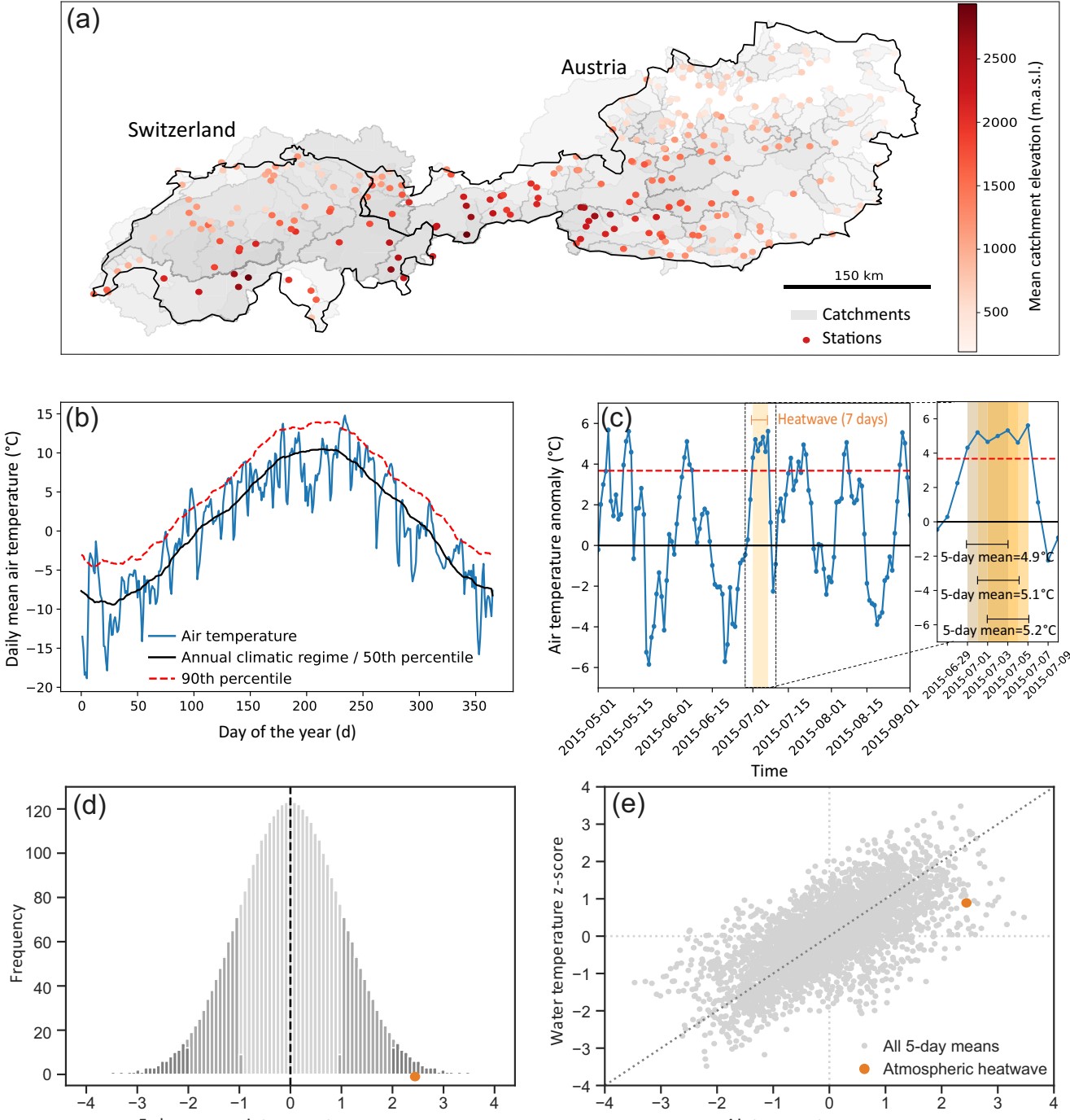

**Fig. 7 | Schematic visualization of the methodology. a** The 275 catchments and measurement stations in Switzerland and Austria. **b** Annual climatic regime and local, seasonally varying 90th percentile threshold for air temperature (for an example station). **c** Air temperature anomalies after removing the annual climatic regime, with one heatwave event where air temperature is above the threshold for more than 5 consecutive days in yellow. The maximum 5-day mean anomaly of the heatwave is 5.2 °C and the average 5-day mean anomaly is 5.07 °C. **d** Standard normal distribution of the 5-day mean air temperature anomalies after transformation to z-scores. The z-score of the heatwave is 2.5, which corresponds to the 98th percentile. **e** Air temperature z-scores plotted against water temperature z-scores. The water temperature z-score during the atmospheric heatwave is approximately 1 (≈ 68th percentile).

for larger catchments, which lies within the minimum duration of an atmospheric heatwave).

### Definition of heatwaves

A heatwave occurs when temperatures show strong positive anomalies for several consecutive days. In line with existing definitions of aquatic and atmospheric heatwaves, we have adopted the following definition for atmospheric and riverine heatwaves in this study: A heatwave is a period during which the daily mean temperature exceeds a local, seasonally varying 90th percentile threshold for a minimum of five consecutive days. The threshold is defined as the 90th percentile of daily mean temperature, centred on a 30-day window and averaged over the 10-year reference period 2011–2021. This definition has been widely used in recent studies to identify riverine heatwaves, including those by Tassone et al.[11,81], Sun et al.[4,16], and Sadayappan and Li[28]. Regarding atmospheric heatwaves, the literature does not provide one single universal definition: the use of a 90th percentile

threshold is common, but some studies focus only on summer heatwaves, while others include humidity to better capture the impact of atmospheric heatwaves on human health[1]. However, winter atmospheric heatwaves can have a substantial impact on mountain hydrology by triggering rapid snowmelt events. Similarly, from the perspective of river ecosystems, riverine heatwaves can be equally relevant to aquatic flora and fauna during both cold and warm months (e.g. for the reproduction of fish species[82,83]). Finally, for the purpose of this study, it is important that both atmospheric and riverine heatwaves are defined in identical ways to facilitate the comparison between the two phenomena, which is why we have selected the above-described definition for both heatwave types.

### Standardization and z-scores

To study the potential link between atmospheric and riverine heatwaves and compare different catchments, seasons, and hydro-climatic driving variables, we standardized all variables to z-scores. A z-score is a statistical measure that quantifies the distance between a data point and the mean of a dataset. Expressed in terms of standard deviations, it indicates where a given value falls within a standard normal distribution and how likely it is that this value occurs.

In a first step, we calculate air temperature anomalies by removing the annual climatic regime from the daily mean air temperature time series (Fig. 7b and c). The annual climatological regime is defined as the 50th percentile of daily mean air temperature, centred on a 30-day window and averaged over the 10-year reference period 2011–2021. Next, we identified atmospheric heatwaves following the definition above (Section 3.2).

Many of the variables of interest have a non-normal distribution, meaning that we cannot calculate z-scores directly to identify the probability of a heatwave. This would make interpreting the z-scores complex and potentially misleading, particularly when dealing with heavily skewed data, such as precipitation data. Instead, we made use of the empirical Weibull probability distribution. The Weibull distribution is recommended when the form of the underlying distribution is unknown and when unbiased exceedance probabilities are desired. For a sample $X$ with population size $n$, the Weibull plotting position (or the probability) of the $j$th element is defined as:

$$(x_j - \alpha)/(n + 1 - \alpha - \beta), \tag{1}$$

with $\alpha = 0$ and $\beta = 0$. To compute the plotting position of a heatwave event using the Weibull distribution of a specific catchment, we first calculated the 5-day mean air temperature anomalies for the entire time series by applying a 5-day centred rolling mean (Fig. 7c). A 5-day window is chosen because heatwaves have a minimum duration of 5 days and we also want to calculate a heatwave-specific plotting position. When heatwaves have a duration of more than 5 days, we calculate the mean of different 5-day mean values within the heatwave duration (Fig. 7c zoom-in). Next, we calculate the Weibull plotting positions (or empirical percentile points) for the 5-day smoothed data per station, including the heatwaves, by using *mstats.plotting_positions* from SciPy. The likelihood of each heatwave can then be presented as a percentile, which represents the rank of the 5-day mean air temperature anomaly of the heatwave event within the Weibull distribution of all data of that station. Finally, we translate the plotting positions (percentiles) via the percent point function (ppf, which is the inverse of cdf) to a z-score in a standard normal distribution (mean = 0, std = 1) by using SciPy *stats.norm.ppf*. This last step is necessary to enable comparison of z-scores between variables and catchments. It results in a time series with z-scores per variable, where every heatwave is given a z-score within the standard normal distribution (Fig. 7d).

For the other variables of interest, we applied the same approach (from calculating 5-day mean anomalies to plotting positions to z-scores), such that we can compare the probability of different variables during the atmospheric heatwaves with the probability of the atmospheric heatwave itself. For example, to define the likelihood of the observed water temperature during a heatwave, we first constructed the time series of the 5-day mean water temperature anomalies and fitted the Weibull distribution to these averaged time series. For each time window for which an atmospheric heatwave occurred, we calculated the plotting position of the average 5-day mean water temperature anomaly in the Weibull distribution and made the transformation to z-scores. Now, we are able to compare the z-score of the atmospheric heatwave with the z-score of water temperature during this heatwave period (Fig. 7e).

Finally, we checked for each atmospheric heatwave whether it overlapped with any riverine heatwave. Each atmospheric heatwave that overlaps with a riverine heatwave, even for just a few days, is labelled as such, regardless of whether the riverine heatwave starts or ends before or after the atmospheric heatwave. Next, we compared the atmospheric heatwaves that co-occurred with riverine heatwaves with those that did not, in order to identify the conditions that strengthen or weaken the development of a riverine heatwave during an atmospheric heatwave.

### Case studies with network perspective

We conducted two case studies to demonstrate the variations in river water temperature sensitivity to atmospheric heatwaves along river networks. The first case study compared the responses of water temperature measured at five different measurement stations along the Drau River (Supplementary Fig. 5) during two similarly severe atmospheric heatwaves. This river and its stations were chosen for the following reasons: the river includes several stations that measure water temperature downstream of each other; the stations are not affected by any large lakes; human regulation is limited; and the catchments include high-elevation areas and a small glacier, which allows us to study the influence of meltwater on riverine heatwave development. The Drau river has its origin at the glaciers of the Großvenediger mountain in the National Park 'Hohe Tauern' (South-eastern Austria). The highest measurement station is located at 1686 m.a.s.l. and the lowest at 555 m.a.s.l, therewith covering an elevation range of 1131 m over a vertical distance of approximately 100 km. The second case study focused on the effect of large lakes on the water temperature response. Specifically, we compared the water temperature z-scores during atmospheric heatwaves directly upstream and downstream of five big lakes in Switzerland, namely Lake Geneva, lake Biel, Lake Brienz, Lake Lucerne, and Lake Walen. These lakes were selected because they are the only lakes for which water temperature stations were available directly upstream and downstream of the lakes. This is important to isolate the lake's influence on warming rather than identifying other processes that can warm the river while the water moves from the lake outlet to the measurement station. More information on the lakes, the regulation of water levels, the hydrological regimes, and the distance from the lake to the upstream and downstream stations is provided in Supplementary Table 1.

### Data availability

The shapefiles of all catchments, including static catchment characteristics, annual regime data and event data are available through HydroShare according to the FAIR data sharing principles: van Hamel, A. (2025)[84].

### Code availability

A python notebook to generate the figures that are provided in this manuscript and in the Supplementary information is available through HydroShare: van Hamel, A. (2025)[84].

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

## Acknowledgements

The authors thank the Swiss Federal Office for the Environment and the Austrian Hydrographic Service for data provision and acknowledge financial support through the Swiss National Science Foundation (Grant 501100001711-218486).

## Author contributions

A.v.H.: conceptualization, formal analyses, writing first draft of manuscript. J.J.: data generation and curation, revision and editing of manuscript. M.B.: conceptualization, revision and editing of manuscript, supervision.

## Competing interests

The authors declare no competing interests.

## Additional information

**Peer review information** : *Communications Earth and Environment* thanks Karsten Schulz and the other, anonymous, reviewer(s) for their contribution to the peer review of this work. Primary Handling Editor: Nicola Colombo. A peer review file is available.

