## [Transparent Peer Review file · Communications Earth & Environment]

River temperature response to atmospheric heatwaves is modulated by discharge and meltwater

Corresponding Author: Professor Manuela Brunner

This manuscript has been previously reviewed at another journal. This document only contains information relating to versions considered at Communications Earth & Environment.

Version 0:

Decision Letter:

Dear Ms Van Hamel,

Your manuscript titled "River temperature response to atmospheric heatwaves is modulated by discharge and meltwater" has now been seen by 3 reviewers, and we include their comments at the end of this message. They find your work of interest, but some important points are raised. We are interested in the possibility of publishing your study in Communications Earth & Environment, but would like to consider your responses to these concerns and assess a revised manuscript before we make a final decision on publication.

In particular, please ensure that the revised manuscript meets the following editorial thresholds:

1. Please present a compelling case that your interpretations and conclusions regarding the mechanism by which riverine heatwaves are triggered by atmospheric heatwaves are robust and fully supported by your analyses.
2. Demonstrate that your statistical approaches are sufficiently robust and reliable to support your conclusions.
3. Ensure that your manuscript demonstrates a clear and significant contribution beyond the scope of existing literature, highlighting its novelty and relevance within the field.

We therefore invite you to revise and resubmit your manuscript, along with a point-by-point response that takes into account the points raised. Please highlight all changes in the manuscript text file.

Please submit your point-by-point responses as a separate file, distinct from your cover letter where you can add responses to the Editors' comments that you do not want to be made available to the reviewers. Word files are preferred. We recommend that any figures, tables or graphs that are included in the response to reviewers are also included in the main article or Supplementary Information.

Please use the following link to submit your revised manuscript, point-by-point response to the referees' comments (which should be in a separate document to any cover letter), a tracked-changes version of the manuscript (as a PDF file) and the completed checklist:

Link Redacted

We hope to receive your revised paper within six weeks; please let us know if you aren't able to submit it within this time so that we can discuss how best to proceed. If we don't hear from you, and the revision process takes significantly longer, we may close your file. In this event, we will still be happy to reconsider your paper at a later date, as long as nothing similar has

been accepted for publication at Communications Earth & Environment or published elsewhere in the meantime.

Please do not hesitate to contact us if you have any questions or would like to discuss these revisions further. We look forward to seeing the revised manuscript and thank you for the opportunity to review your work.

Best regards,

Nicola Colombo, PhD
Associate Editor, Communications Earth & Environment
Consulting Editor, Communications Sustainability

EDITORIAL POLICIES AND FORMATTING

- Behavioural and social science
- Ecological, evolutionary & environmental sciences
- Life sciences

Furthermore, please align your manuscript with our format requirements, which are summarized on the following checklist: <https://www.nature.com/documents/commsj-phys-style-formatting-checklist-article.pdf> Communications Earth & Environment formatting checklist

and also in our style and formatting guide <https://www.nature.com/documents/commsj-phys-style-formatting-guide-accept.pdf> Communications Earth & Environment formatting guide .

*** DATA: Communications Earth & Environment endorses the principles of the Enabling FAIR data project (<http://www.copdess.org/enabling-fair-data-project/>). We ask authors to make the data that support their conclusions available in permanent, publically accessible data repositories. (Please contact the editor if you are unable to make your data available).

All Communications Earth & Environment manuscripts must include a section titled "Data Availability" at the end of the Methods section or main text (if no Methods). More information on this policy, is available at <http://www.nature.com/authors/policies/data/data-availability-statements-data-citations.pdf>.

If a community resource is unavailable, data can be submitted to generalist repositories such as <https://figshare.com/> or <http://datadryad.org/> Dryad Digital Repository. Please provide a unique identifier for the data (for example a DOI or a permanent URL) in the data availability statement, if possible. If the repository does not provide identifiers, we encourage authors to supply the search terms that will return the data. For data that have been obtained from publically available sources, please provide a URL and the specific data product name in the data availability statement. Data with a DOI should be further cited in the methods reference section.

REVIEWER COMMENTS:

Reviewer #1 (Remarks to the Author):

Review

This well written paper addresses an important topic regarding water temperature and heat waves based on case studies from the Swiss and Austrian Alps. The statistical analyses are correct, the figures of good quality.

The overall hypothesis is not new, previous work has addressed that question in a similar region (Piccolroaz et al., 2016; Piccolroaz et al., 2018; Toffolon and Piccolroaz, 2015; Michel et al., 2020; Michel et al., 2022) (“We hypothesize that the origin and temperature of the water feeding a stream are probably as relevant as the amount of discharge (and the river’s thermal capacity) in controlling river water temperature.”). The actual novelty becomes clear a few lines later and focuses on heat waves and not on general water temperature processes (“However, they did not consider other hydro-climatic variables, such as meltwater inflow, relative humidity or drought indices. »). The hypothesis could thus be better framed.

I am not sure how valuable the additional focus on relative humidity, soil moisture, radiation and meteorological water availability is without discussing how these variables co-vary with air temperature and discharge / melt water (i.e. how relevant is it to investigate statistical distributions of variables that already influence the main drivers?). This would at least need a clear discussion for the non-expert readers. Furthermore: if meltwater contribution is explicitly investigated, groundwater contribution should receive the same focus: groundwater is a key source of heat input or absorbance for rivers, but it receives almost no attention here. For the large low-land catchments, this is a limitation

Overall, the regional analysis is based on conditional z-score distributions (conditional on having an atmospheric heat wave or not). While this is simple and thus appealing, I am not convinced that discussing positive and negative excursions of z-scores is a robust enough metric to evaluate the co-variation between variables.

Furthermore, the present work does not propose a classification of the rivers according to their hydrologic regime and degree of regulation (in particular hydropower, lake regulation) (e.g. Piccolroaz et al., 2016). Even if water regulation is mentioned, the fact that rivers are not explicitly classified into regulated and not regulated is a problem. E.g. how many rivers above 2000 m asl are actually not regulated and represent natural patterns? Furthermore, the hydrological regime and the distance e.g. to glacier should be reported; as far as I see, e.g. the work of (Gallice et al., 2015) focusing on natural regimes only reported a single water temperature measurement station that was influenced by glacier water in Switzerland.

Overall, I feel that the paper offers a very simplistic view on rivers. E.g. the occurrence of winter heat waves in snow-dominated systems is explained by mid-winter rainfall effects or groundwater inflow, ignoring thereby that winter water temperatures at high elevations are strongly driven by atmospheric exchange along the stream network. In general, the interplay of in-stream processes versus hillslope or groundwater inflow is not discussed explicitly. This discussion would require stratifying the measurement locations according e.g. to the Strahler order. Accordingly, I tend to not agree with the statement “With this work, however, we contribute to a growing understanding of these complex dynamics, specifically the influence of discharge and meltwater on the sensitivity of river water temperature to heatwaves. » (line 233)

Regarding the case study along the stream: the results are certainly noteworthy but simply confirm existing knowledge for two heatwaves.

The lake case is analyzed superficially: we do not know how far the stations are from the lakes – is the water temperature there a mere signal of the lake temperature or did the river water warm up between the lake outlet and the measurement station? Which lakes are regulated, which ones have natural outflow, what is the hydrologic regime upstream / downstream? How can 5 different lakes contribute any physical insight? I recommend removing this case study or give more details.

Overall, I am not sure if this study deserves publication in the intended journal: the case study is certainly interesting but little specific new knowledge is presented.

Detailed comments

- How does your definition of heatwave compare to the one of other authors? Given that your definition allows heatwaves to occur in winter, could this not be misleading (terminology-wise) in comparison to previous work?
- Is there a difference between “availability of meltwater » and the « meltwater fraction of the runoff that potentially reaches the river » (see lines 139 and 140)
- Is the meteorological data set useful for the smaller catchments in areas with high topography? What percentage of the catchments are headwater catchments or catchments smaller e.g. than 300 km², for which the resolution of the meteo data is not good enough?
- Same question for the gridded daily snowmelt and glacier : what is the model resolution at the study latitude? And is it reliable for small catchments?
- The z-scoring procedure for precipitation and snowmelt is clear (line 370) but it is unclear how positive or negative z-scores during heatwaves can be interpreted in this case since the probability of precip being zero during the heatwaves is not part of that metric

Gallice, A., Schaefli, B., Lehning, M., Parlange, M. B., and Huwald, H.: Stream temperature prediction in ungauged basins: review of recent approaches and description of a new physics-derived statistical model, *Hydrol. Earth Syst. Sci.*, 19, 3727-3753, 10.5194/hess-19-3727-2015, 2015.

Michel, A., Brauchli, T., Lehning, M., Schaefli, B., and Huwald, H.: Stream temperature and discharge evolution in Switzerland over the last 50 years: annual and seasonal behaviour, *Hydrology and Earth System Sciences*, 24, 115-142, 10.5194/hess-24-115-2020, 2020.

Michel, A., Schaefli, B., Wever, N., Zekollari, H., Lehning, M., and Huwald, H.: Future water temperature of rivers in Switzerland under climate change investigated with physics-based models, *Hydrol. Earth Syst. Sci.* 2021, 1-45, 10.5194/hess-26-1063-2022, 2022.

Piccolroaz, S., Calamita, E., Majone, B., Gallice, A., Siviglia, A., and Toffolon, M.: Prediction of river water temperature: a comparison between a new family of hybrid models and statistical approaches, *Hydrological Processes*, 30, 3901– 3917, 10.1002/hyp.10913, 2016.

Piccolroaz, S., Toffolon, M., Robinson, C. T., and Siviglia, A.: Exploring and Quantifying River Thermal Response to Heatwaves, *Water*, 10, 1098, 2018.

Toffolon, M., and Piccolroaz, S.: A hybrid model for river water temperature as a function of air temperature and discharge, *Environmental Research Letters*, 10, 114011, 2015.

Reviewer #2 (Remarks to the Author):

Review of the manuscript:

This manuscript uses high-frequency observational datasets to investigate the response of riverine heatwaves to atmospheric heatwaves under different hydro-climatic conditions. The authors report that only 47% of atmospheric heatwaves lead to riverine heatwaves, and that the occurrence of riverine heatwaves often requires not only atmospheric forcing but also additional hydro-climatic conditions such as anomalous discharge and meltwater. Furthermore, the study presents a detailed case analysis of a river network to explore the interactions among riverine heatwaves, atmospheric heatwaves, and hydro-climatic variables.

Compared with previous studies, which primarily focused on long-term changes in riverine heatwave characteristics such as intensity, frequency and duration, then tended to attribute them in a simplified manner to atmospheric heatwaves, rising air temperatures, or discharge anomalies, this study provides a more systematic perspective. Specifically, it analyzes the hydro-climatic conditions under which atmospheric heatwaves are likely to trigger riverine heatwaves, thereby offering valuable insights into the driving mechanisms of riverine heatwaves.

While the manuscript addresses an important and timely topic, several aspects require clarification and further development: 1. The central question for me is how the authors define that riverine heatwaves are triggered by atmospheric heatwaves. Is this based solely on temporal overlap? If so, I am not convinced this is a robust approach, this only reflects coincidence or probability rather than causality. Recent studies (Wang et al., 2024a, b; Sun et al., 2025) have used model simulations and controlled experiments to explicitly determine the role of atmospheric heatwaves in driving freshwater heatwaves. The authors should clarify their criteria and consider whether this approach is robust, because it is the base and core of this manuscript.

2. Line 87-88: The manuscript reports that the link between air and water temperatures is strongest in catchments at low elevations with a small snow fraction. Although Figure 1c seems to suggest this pattern, the figure needs added relationship value and statistical significance to support the conclusion.

3. Line 110: The suitability of the term “decoupling” is questionable. Previous studies (e.g., Cassie., 2006) have shown that air and water temperatures follow an S-shaped relationship: linear in the mid-range but nonlinear at low and high extremes, where river water temperature ceases to increase proportionally with air temperature. Since riverine heatwaves are extreme events, it is unlikely that these processes represent a true “decoupling” between atmospheric and riverine heatwaves. The authors should reconsider their terminology and provide a more accurate interpretation.

4. Line 127: The manuscript emphasizes cases where atmospheric heatwaves coincide with riverine heatwaves. However, it would also be valuable to examine cases where riverine heatwaves occur in the absence of atmospheric heatwaves. Do such cases exist? If so, comparing these with the co-occurrence cases could provide important insights into additional hydro-meteorological influences.

5. Line 154: The described phenomenon is puzzling. In general, solar radiation is an important driver of river water temperature, while high humidity, which as authors described often associated with rainfall, reduces solar radiation reaching the water surface and thereby suppresses warming. In the discussion, the authors suggest that high humidity and reduced radiation accompany warmer temperatures, rainy, or stormy weather, which may lead to riverine heatwaves. However, it is unclear whether rainy or stormy conditions trigger or suppress riverine heatwaves, and I am not aware of references supporting this interpretation. A more detailed analysis, supported by relevant literature, is needed to provide a convincing explanation of this mechanism.

6. Line 159: Why were these five stations in Drau River chosen? Were they the only locations where this phenomenon was observed, or were there other criteria guiding the selection? A more transparent justification would be helpful.

Overall, while the study provides valuable insights, these clarifications are necessary before the manuscript can be considered for publication.

Reference:

Wang, W. et al. The impact of extreme heat on lake warming in China. *Nat Commun* 15, 70 (2024).

Wang, X. et al. Disproportionate impact of atmospheric heat events on lake surface water temperature increases. *Nat. Clim. Chang.* (2024) doi:10.1038/s41558-024-02122-y.

Sun, J. et al. Impact of extreme atmospheric heat events on river thermal dynamics and heatwaves. *Journal of Hydrology* 659, 133292 (2025).

Caissie, D., 2006. The thermal regime of rivers: a review. *Freshw. Biol.* 51 (8), 1389-1406.

Reviewer #3 (Remarks to the Author):

Review of manuscript COMMSENV-25-4311

„River temperature response to atmospheric heatwaves is modulated by discharge and meltwater”

I. Summary of the most important scientific findings

- (1) This paper addresses a very important problem of analyzing the potential for mitigation of atmospheric heatwaves in river systems due to additional meltwater distribution. In particular, river runoff regimes will be highly affected under climate change conditions, while in parallel snow lines are moving to larger altitudes and glaciers are significantly retreating.
- (2) For a very large data set in Austrian Alps including longer term data (2011-2021) from 275 catchments in Austria and Switzerland, atmospheric heat waves as well as riverine heat waves were identified and analyzed.
- (3) Their results demonstrated that the river temperature response can be very different, depending on the location of the catchment/temperature measurement. Only for 47% of the heatwaves analyzed lead to riverine heatwaves.
- (4) This was controlled to a large extent by discharge level and melt water contributions, with high discharge and high meltwater contributions preventing riverine heatwaves, while a strong link exists between low discharge and no-meltwater contributions and the generation of riverine heat waves.

II. Novelty and contributions of this paper

- (5) While such relationships/effects between air temperature and river temperature (or atmospheric and riverine heatwaves) have been found investigated previously (but only using smaller number of catchments and/or using modelling results), I believe this study is of very high value to the scientific community as it is based on a large number of locations in a very heterogeneous and climate sensitive area in Central Europe. In particular, I like that this study is based measured/observed discharge & temperature data.
- (6) Also, making these available to larger scientific community is a very valuable contribution to a large number of research institutes

III. Critical comments and recommendations to the editor

- (7) I have very much enjoyed reading this manuscript. It is highly relevant to hydrologist with some focus on climate change related topics in water resource management or ecology. However, there are two may be more critical questions or concerns I have that are listed in the following:
 - (8) How did you deal with the time lag between temperature induced melting and runoff routing to the measurement site – maybe I missed such a statement.
 - (9) You used PCR-GLOBEWB as a hydrological model that to my knowledge uses a Day-Degree method for modelling melting processes. We see in our analysis that there is a significant difference to more complex radiation driven approaches especially during melting – can you comment on how this would affect the analysis in your study. I guess commenting on this in the method section would be needed.I have a number of minor/specific comments that should be relatively easy to address, so that I would recommend to the editor to accept the manuscript after some revisions.

IV. Minor and specific comments

- L15: I would already define heatwave here early in the introduction – you kept me asking this during the next lines until it appears in L64ff
- L87: Define AT and WT.
- L108: potentially sounds pretty vague!
- L154: I would be interested in why high humidity and rainy whether favor heat waves?
- L240: I would argue that heat capacity of (lake) water is a physical constant and cannot be influenced
- L308: PCR-GLBWB is a model that to my knowledge

** Visit Nature Portfolio's author and referees' website at www.nature.com/authors for information about policies, services and author benefits**

Communications Earth & Environment is committed to improving transparency in authorship. As part of our efforts in this direction, we are now requesting that all authors identified as 'corresponding author' create and link their Open Researcher and Contributor Identifier (ORCID) with their account on the Manuscript Tracking System prior to acceptance. ORCID helps the scientific community achieve unambiguous attribution of all scholarly contributions. You can create and link your ORCID from the home page of the Manuscript Tracking System by clicking on 'Modify my Springer Nature account' and following the instructions in the link below. Please also inform all co-authors that they can add their ORCIDs to their accounts and that they must do so prior to acceptance.

If you experience problems in linking your ORCID, please contact the Platform Support Helpdesk.

Version 1:

Decision Letter:

Dear Dr. Brunner,

Your manuscript titled "River temperature response to atmospheric heatwaves is modulated by discharge and meltwater" has now been seen by our reviewers, whose comments appear below. In light of their advice we are delighted to say that we are happy, in principle, to publish a suitably revised version in Communications Earth & Environment, provided you present a clear discussion of the limitations of your approach, including the limited representation of the influence of groundwater.

We therefore invite you to revise your paper one last time to address the remaining concerns of our reviewers. At the same time we ask that you edit your manuscript to comply with our format requirements and to maximise the accessibility and therefore the impact of your work.

EDITORIAL REQUESTS:

****Please take care to match our formatting and policy requirements. We will check revised manuscript and return manuscripts that do not comply. Such requests will lead to delays. ****

SUBMISSION INFORMATION:

OPEN ACCESS:

Communications Earth & Environment is a fully open access journal. Articles are made freely accessible on publication. For further information about article processing charges, open access funding, and advice and support from Nature Portfolio, please visit <https://www.nature.com/commsenv/open-access>

Link Redacted

Best regards,

Nicola Colombo, PhD
Associate Editor, Communications Earth & Environment
Consulting Editor, Communications Sustainability

REVIEWERS' COMMENTS:

Reviewer #1 (Remarks to the Author):

I thank the authors for their answers and the detailed revisions, the manuscript has improved even if no complementary analysis was performed by the authors. I just would like to add here that I do not agree with the view that soil moisture or

atmospheric water availability are proxies for groundwater droughts: hydrologic droughts are different from soil moisture or atmospheric droughts. In absence of groundwater data, an actual proxy for groundwater could e.g be derived from streamflow (e.g. based on a baseflow analysis). A final version should certainly be even more explicit on the fact that the influence of groundwater is not well represented in this analysis.

Reviewer #2 (Remarks to the Author):

Review of the manuscript:

The authors have carefully responded to the comments raised in the first round of review, and the revisions and responses are generally clear and effective in addressing my previous concerns. The overall quality of the manuscript has improved substantially, and the revised version is clearly stronger than the original submission. I also appreciate the authors' decision to rely on in situ observations, which represents a clear strength of this study. Given the limited number of studies that investigate riverine heatwaves based on observed river temperature data, this observational perspective provides an important and valuable contribution to the field.

The methodology adopted in the manuscript primarily identifies potential "triggering" mechanisms based on the temporal overlap between atmospheric and riverine heatwaves. This framework appears to focus more on describing compound or co-occurring heatwave events rather than establishing a clear causal relationship. In addition, the thermal response of river systems often exhibits pronounced lag effects due to hydrological and geomorphological controls, which may not be fully captured by simple temporal concurrence analyses. This point was also raised in my first-round review. At the same time, even if this approach may currently represent the best feasible methodology based on available observational datasets, its robustness and inherent limitations still warrant careful discussion. While the authors argue that this observation-based approach is preferable to previous modeling studies. I also acknowledged that numerical models, despite introducing their own sources of uncertainty, remain the most direct method for isolating atmospheric forcing through controlled, variable-specific experiments.

Therefore, I strongly recommend that the authors include a dedicated paragraph or section to introduce the limitations of the current methodology, as well as a balanced comparison of the strengths and weaknesses of this study approach relative to modeling-based methods used in previous studies.

Reviewer #3 (Remarks to the Author):

The authors have addressed my comments very well, and I am more than happy to recommend the paper for publications. I have also read through the other colleagues comments, and I also think the authors have done an excellent job in addressing them.

The overall manuscript has now become even stronger and I am looking forward to read it very soon in "Communications - Earth and Environment"

** Visit Nature Portfolio's author and referees' website at www.nature.com/authors for information about policies, services and author benefits**

Point-by-point reply to the comments of the three reviewers

Thank you very much for giving us the opportunity to revise our manuscript titled *‘River temperature response to atmospheric heatwaves is modulated by discharge and meltwater’*. We highly appreciate the constructive comments and specific suggestions by the three reviewers on how to improve our manuscript. Below, we provide you with the point-by-point reply to the reviewers’ comments, with their comments in **black** and our replies in **blue**. Please note: Newly added text is written in *italics*. For each adjustment, we include the line of text in the manuscript where this change is made (see the manuscript version with changes marked in **red**).

Reviewer 1:

Review

This well written paper addresses an important topic regarding water temperature and heat waves based on case studies from the Swiss and Austrian Alps. The statistical analyses are correct, the figures of good quality.

Comment: The overall hypothesis is not new, previous work has addressed that question in a similar region (Piccolroaz et al., 2016; Piccolroaz et al., 2018; Toffolon and Piccolroaz, 2015; Michel et al., 2020; Michel et al., 2022) (“We hypothesize that the origin and temperature of the water feeding a stream are probably as relevant as the amount of discharge (and the river’s thermal capacity) in controlling river water temperature.”). The actual novelty becomes clear a few lines later and focuses on heat waves and not on general water temperature processes (“However, they did not consider other hydro-climatic variables, such as meltwater inflow, relative humidity or drought indices. »). The hypothesis could thus be better framed.

Reply: Thank you for your comment. We have made some adjustments to the introduction to clarify and better frame the knowledge gaps, our hypothesis, and the novelty of our work. Most importantly, we have added the following text to the paragraph where we discuss the research gaps (Line 61-67):

“[...] However, they did not consider other hydro-climatic variables, such as meltwater inflow, relative humidity or drought indices. As a result, we still lack a detailed understanding of the main processes that control the link between atmospheric and riverine heatwaves, especially in mountainous regions and across different seasons.

In this study, we test the hypothesis that water temperature is not always following air temperature extremes and that other processes, such as hydrological conditions (e.g. inflow of groundwater and meltwater) and atmospheric conditions (e.g. humidity and radiation), might also play an important role in mitigating or enhancing river warming during atmospheric heatwaves. In doing so, we aim to improve the understanding of how river water temperature responds to atmospheric heatwaves. [...]”

Comment: I am not sure how valuable the additional focus on relative humidity, soil moisture, radiation and meteorological water availability is without discussing how these variables co-vary with air temperature and discharge / melt water (i.e. how relevant is it to investigate statistical distributions of variables that already influence the main drivers?). This would at least need a clear

discussion for the non-expert readers. Furthermore: if meltwater contribution is explicitly investigated, groundwater contribution should receive the same focus: groundwater is a key source of heat input or absorbance for rivers, but it receives almost no attention here. For the large low-land catchments, this is a limitation.

Reply: Thank you for highlighting your doubts about the usefulness of the wide set of hydro-climatic variables that we included in our analysis. Apparently, we were not able to communicate their value in the manuscript. In short, we believe that it is actually very valuable to analyze such a wide range of hydro-climatic variables since our hypothesis is that (in certain rivers) water temperature will not directly react to air temperature during atmospheric heatwaves, because other processes might have an influence as well and mitigate river warming. To clarify this better in our manuscript, we have made some adjustments to the third paragraph of our introduction where we discuss the importance of other variables than air temperature on controlling water temperature (Line 36-45):

“[...] In small (headwater) rivers in particular, air temperature is not always an accurate indicator of river water temperature, because local factors such as groundwater inflow and outflow, soil moisture, snow water equivalent, impoundments, stream size, and shading by riparian vegetation can play an important role too (van Hamel et al., 2024; Beaufort et al., 2022; Fellman et al., 2014; Caissie, 2006; Moatar and Gailhard, 2006). Substantial groundwater or meltwater inflow can potentially mitigate the effect of high air temperatures on river warming (Moore et al., 2009; Piccolroaz et al., 2018), while low discharges during drought can reduce the heat capacity of rivers and make them more prone to warming (Caissie, 2006; White et al., 2023). Furthermore, while air temperature is typically used as an indicator of heat exchange at the air–water interface, other drivers such as radiation and evaporative cooling also become important as air temperature increases (Caissie, 2006). Therefore, a comprehensive understanding of the response of water temperature requires consideration of a wide range of hydroclimatic variables that might influence the link between air and water temperature.”

Furthermore, we agree with your comment that groundwater is a key variable that can control river water temperature. Unfortunately, however, there is limited observational data available for groundwater, especially for a large-sample catchment study as we present here. One option would have been to include data from groundwater wells; however, single wells are mostly not representative at the catchment scale and the spatial density of this type of data is generally not great, with a very limited number of monitored wells at high elevations. Another option would have been to include modelled groundwater data, but this also introduces large uncertainty because model performance can't be properly evaluated on the sparse observations. Instead of using not very reliable groundwater data (observational or model-based), we therefore decided to focus on drought indicators that we could extract directly from the reanalysis CERRA data set, which also allows for consistency throughout our analysis. Low soil moisture and atmospheric water availability (precipitation minus evaporation) are used as indicators for dry conditions associated with low groundwater levels and potentially less groundwater inflow to the river system.

Based on your comments, we've made some textual adjustments to the discussion section (see Line 248-261). With that we hope to better reflect the potential importance of groundwater, how we made use of the drought indicators in our analysis, and how our results align with exiting studies:

“Low flow anomalies in summer and spring are often associated with dry periods, which include high atmospheric energy inputs, little water availability with precipitation deficits and excessive evaporation, and low groundwater levels (White et al., 2023; Bruno et al., 2022). *This reduction in groundwater levels and stream-groundwater connectivity can make rivers more sensitive to atmospheric warming (Hare et al., 2021; Tague et al., 2007) and consequently favour the development of riverine heatwaves. However, in contrast to our expectations, we found no clear relationship between riverine heatwave development and the selected drought indices, such as water availability (i.e. precipitation minus evaporation) and soil moisture. This could be related to the delayed influence of these variables on river water temperature. Similarly, some studies have shown clear differences in river water temperature between drought and non-drought years, while others reported minor or negligible changes, likely due to the complex, river-dependent interplay of energy flux dynamics, reach-scale habitat conditions, and water source contributions, which haven't been fully understood yet (White et al., 2023). Ideally, observational or modelled groundwater data would have been used instead of drought indices to better capture these dynamics. However, reliable baseflow estimates that have been evaluated against observations remain challenging to obtain across large spatial domains due to a lack of adequate observational data.*”

Comment: Overall, the regional analysis is based on conditional z-score distributions (conditional on having an atmospheric heat wave or not). While this is simple and thus appealing, I am not convinced that discussing positive and negative excursions of z-scores is a robust enough metric to evaluate the co-variation between variables.

Reply: Our analysis, which is based on comparing z-scores of different variables during the time window of atmospheric heatwave occurrence, makes use of a *co-occurrence perspective*. Please note that co-occurrence is not the same as co-variation and that we do not intend to prove co-variation between variables. With the co-occurrence approach, we study the association between two variables. By studying how different variables behave during a specific time window, we can reveal potential underlying patterns and relationships between the two variables. Co-occurrence analyses of different variables have been used in many other hydrological studies, e.g. to understand the linkage between wildfires and atmospheric and marine heatwaves (Santos et al., 2024), or to understand the compound occurrence of heatwaves, meteorological and hydrological droughts, and floods (Vieira Passos et al., 2024). We consider the co-occurrence approach to be useful to study the relationship between atmospheric and riverine heatwaves because it can be used as an association metric and allows us to test certain hypotheses.

In our study, we hypothesize that normal conditions in the drivers (in line with the climatological mean) will lead to normal water temperatures, whereas the development of extreme water temperatures will likely necessitate an offset from normal conditions in one or multiple drivers. To test this hypothesis, we compare the 'extremeness' of the atmospheric heatwaves with the 'extremeness' of the water temperature response. In this comparison, we also consider how other driving variables, such as snowmelt, discharge, soil moisture, and radiation, behave, in order to account for compounding effects. To test for differences in the generation processes of atmospheric heatwave leading to riverine heatwaves and those that don't, we created two sub-samples of reasonable size by differentiating between atmospheric heatwaves that co-occurred with/without riverine heatwaves. We showed that there is indeed a significant difference between the hydro-climatic conditions associated with the two event samples (statistical significance is proved by using

the Mann-Whitney U test). This gives us sufficient confidence to draw conclusions on the association between variables based on this comparison.

- Santos, R., Russo, A. & Gouveia, C.M. Co-occurrence of marine and atmospheric heatwaves with drought conditions and fire activity in the Mediterranean region. *Scientific Reports* 14, 19233 (2024). <https://doi.org/10.1038/s41598-024-69691-y>
- Vieira Passos, M., Kan, J.C., Destouni, G. et al. Identifying regional hotspots of heatwaves, droughts, floods, and their co-occurrences. *Stoch Environ Res Risk Assess* 38, 3875–3893 (2024). <https://doi.org/10.1007/s00477-024-02783-3>

Comment: Furthermore, the present work does not propose a classification of the rivers according to their hydrologic regime and degree of regulation (in particular hydropower, lake regulation) (e.g. Piccolroaz et al., 2016). Even if water regulation is mentioned, the fact that rivers are not explicitly classified into regulated and not regulated is a problem. E.g. how many rivers above 2000 m asl are actually not regulated and represent natural patterns? Furthermore, the hydrological regime and the distance e.g. to glacier should be reported; as far as I see, e.g. the work of (Gallice et al., 2015) focusing on natural regimes only reported a single water temperature measurement station that was influenced by glacier water in Switzerland.

Reply: A classification of the rivers according to their hydrological regime (e.g. nival vs pluvial regime) does not add much additional information to our analysis, as we already take into account other catchment characteristics such as catchment elevation and the ‘percentage of precipitation falling as snow’. These characteristics can be used as a proxy for the hydrological regime: our analysis shows that in catchments with high mean elevation and a high ‘percentage of precipitation falling as snow’ fewer atmospheric heatwaves manage to translate to riverine heatwaves compared to catchments with a low elevation and a low ‘percentage of precipitation falling as snow’ (also see Figure 1 in our manuscript). A similar result is observed when we split the catchments based on their hydrological regime, where catchments with a nival regime result in a smaller percentage of atmospheric heatwaves that translate into riverine heatwaves compared to catchments with a mixed or pluvial regime: for catchment with a nival regime, 37% of the atmospheric heatwaves co-occurs with riverine heatwaves (64% does not), while for catchments with a pluvial regime, 62% of the atmospheric heatwaves co-occurs with riverine heatwaves (and 38% does not).

Nonetheless, based on your comment, we have decided to add the information about the hydrological regime to the overview with catchment characteristics, which will be made accessible via Hydroshare. The hydrological regime of each catchment is defined based on the Pardé coefficient and hierarchical clustering: The Pardé coefficient describes a river's seasonal flow patterns by comparing its mean monthly flow to its mean annual flow. Hierarchical clustering (with the Euclidean distance matrix and the Ward variance) is used to obtain three clusters based on the Pardé coefficients, which reflect three different hydrological regimes: nival, pluvial, and mixed. The result is shown below:

We considered classifying the catchments based on the level of human regulation. However, we concluded that this would not add value for the following two reasons:

- First, consistent information on the degree of regulation is unfortunately not available for the large number of catchments that we included in our study. The main issue is that different countries and datasets apply different categories and methods to define the degree of regulation which makes it difficult to assess the degree of regulation in a collective and comparable way: in Switzerland the human influence on river discharge is described based on 7 categories (water regulation, hydropower, wastewater, etc.) where each category receives a score between 0-1 (Steeb et al, 2024). In Austria, however, LamaH-CE (Klingler et al., 2021) provides information on the degree of regulation based on a very different classification, which distinguishes between 13 types of human impact (water withdrawals, water intakes, infiltration, reservoirs, etc.) with 4 levels for the degree of impact (no, small, moderate, and strong impact). Furthermore, detailed information on how far upstream of the measurement station such impacts/disturbances are located is often not available and would require a very labor-intensive and catchment-specific investigation.
- Second, we checked whether splitting between human influenced and natural catchments would affect our results. Although it was not possible to make such a split for the entire dataset (for the reasons just explained above), we did make a split for all Austrian catchments. For catchments with no or very low human impact (n=111), 45% of the atmospheric heatwaves co-occurred with riverine heatwaves, while for catchments with significant human impact (n=96) 48% of the atmospheric heatwaves co-occurred with a riverine heatwave. Based on this minor difference, we concluded that elevation, percentage of precipitation falling as snow, and the hydrological regime have a much larger influence on the sensitivity of water temperature to atmospheric heatwaves than the level of regulation.

Taking the above points into consideration, we believe it is important to explain to readers why we did not pay more attention to the impact of human influences and regulations in our research. Therefore, we have added a few sentences on this in the discussion section (Line 278-282):

“[...] In addition, this study did not specifically account for human influences, such as the effect of reservoir regulation, water withdrawals, and thermal effluents from industries. Quantifying the effect of human influences on the relationship between air and water temperatures is challenging because of a lack of consistent information on regulation, particularly across the large, cross-border sample of

catchments considered in our study. We recommend that future large-sample studies explicitly investigate and consider human influences.”

Finally, we have extended the description of catchment characteristics by sharing information about the mean daily specific discharge (mm d^{-1}), the glacier coverage per catchment (%), the percentage of precipitation that falls as snow (%), and the hydrological regimes (-). This information, in addition to other catchment characteristics, such as the catchment area and elevation (mean, median, and range), will be made available upon publication through Hydroshare according to the FAIR data sharing principles.

- Steeb N., Lustenberger F., Zappa M. (2024). Beurteilung der Beeinflussung des Abflusses an NAWAMessstellen. Detailbericht des BAFU-Projekts HydCheck. Eidg. Forschungsanstalt WSL, Birmensdorf, 69 S, <http://doi.org/10.55419/wsl:37799>.

Comment: Overall, I feel that the paper offers a very simplistic view on rivers. E.g. the occurrence of winter heat waves in snow-dominated systems is explained by mid-winter rainfall effects or groundwater inflow, ignoring thereby that winter water temperatures at high elevations are strongly driven by atmospheric exchange along the stream network. In general, the interplay of in-stream processes versus hillslope or groundwater inflow is not discussed explicitly. This discussion would require stratifying the measurement locations according e.g. to the Strahler order. Accordingly, I tend to not agree with the statement “With this work, however, we contribute to a growing understanding of these complex dynamics, specifically the influence of discharge and meltwater on the sensitivity of river water temperature to heatwaves. » (line 233)

Reply: Based on your comment, we have extended one of the paragraphs of the discussion section where we discuss the heat exchange at different interfaces (e.g. atmosphere-surface water and riverbed-river interface) and in relation to hillslope and in-stream processes (Line 271-278):

“However, even with some understanding of these processes, disentangling the influence of in-stream processes (e.g. streambed friction, turbulence, and river heat capacity) and local conditions around the stream remains challenging and it is yet unclear which of the two has a larger influence. By analysing the simultaneous anomalies in hydro-climatic conditions, we provide some insight into the processes at the atmosphere-water surface interface (e.g. radiation, humidity), the riverbed-river interface (e.g. heat conduction via groundwater in- or outflow), and the hillslope (e.g. surface runoff of precipitation and/or meltwater) that can weaken the thermal sensitivity of water to air temperature, but we are still not able to reflect the full range of processes and their interactions.”

The statement “With this work, ... to heatwaves” (formerly at line 233), has been removed due to the reorganization of paragraphs in the discussion section (see also earlier comments).

Comment: Regarding the case study along the stream: the results are certainly noteworthy but simply confirm existing knowledge for two heatwaves.

The lake case is analyzed superficially: we do not know how far the stations are from the lakes – is the water temperature there a mere signal of the lake temperature or did the river water warm up between the lake outlet and the measurement station? Which lakes are regulated, which ones have natural outflow, what is the hydrologic regime upstream / downstream? How can 5 different lakes contribute any physical insight? I recommend removing this case study or give more details.

Reply: We believe that the use of case studies adds value to our study because it helps to connect and translate our large-scale observations to location-specific conditions. Making the translation to a network approach can be especially insightful for the interdisciplinary audience of *Communications Earth & Environment*, because this perspective brings together knowledge from different disciplines (e.g. the snow, river, and lake communities).

However, we agree that the case studies can be strengthened by providing additional background information. The five lakes were selected because they are the only ones in our study area for which we have water temperature observations in close proximity upstream and downstream of the lake. The downstream stations are mostly located directly at the lake outlet or up to 6 km downstream. Therefore, we think that it is legitimate to assume that the measured water temperatures at the downstream stations provide a realistic representation of the temperature at the lake and its outlet. Furthermore, only one of the five lakes (Lake Walen) has a completely natural outflow. The other four lakes are regulated at/close to the lake outlet to control the water level in the event of exceptionally high or low water levels.

In the Methods section, we have added a few sentences to explain why we selected these lakes and where additional information on the lakes can be found (Line 430-438):

“The second case study focused on the effect of large lakes on the water temperature response. Specifically, we compared the water temperature z-scores during atmospheric heatwaves directly upstream and downstream of five big lakes in Switzerland, namely Lake Geneva, Lake Biel, Lake Brienz, Lake Luzern, and Lake Walen. *These lakes were selected because they are the only lakes for which water temperature stations were available directly upstream and downstream of the lakes. This is important to isolate the lake's influence on warming rather than identifying other processes that can warm the river while the water moves from the lake outlet to the measurement station. More information on the lakes, the regulation of water levels, the hydrological regime, and the distance from the lake to the upstream and downstream stations is provided in Supplementary Table 1.*”

In addition, we have added the following table (Table 1) to the supplementary material, in which we provide more information on the lake area, type of regulation, the hydrological regime, and the distance of the upstream and downstream stations to the lake.

Table 1. Additional details on the 5 lakes that have been selected for the case study. This table is added to the supplementary information that supports our manuscript.

Lake Biel	
Lake surface area	39 km ²
Natural/ regulated	Regulation of the water levels of Lake Biel is controlled in coordination with those of Lake Neuenburger and Lake Murten. The regulation allows for an annual cycle with seasonal variations in water levels. Average discharge at the lake outlet: 241 m ³ /s. Highest measured discharge: 761 m ³ /s.
Distance from upstream station	0.8 km
Distance to downstream station	4.5 km
Hydrological regime	Nival regime (at both the up- and downstream station)
Lake Brienz	
Lake surface area	29.8 km ²
Natural/ regulated	Regulation of the lake outflow 2 km downstream of lake outlet. The outflow of lake Brienz flows through two small hydropower plants (Mühle Burgholz and Livta), and when discharge exceeds 26 m ³ /s also through the gates of one big

	weir. Another smaller weir can be opened during exceptionally high outflows. Lake regulation remains limited to periods of exceptionally high water levels. Average discharge at the lake outlet: 62 m ³ /s. Highest measured discharge: 344 m ³ /s.
Distance from upstream station	3.4 km
Distance to downstream station	0 km, directly at lake outlet and upstream of regulation structures
Hydrological regime	Nival regime (at both the up- and downstream station)
Lake Geneva	
Lake surface area	581 km ²
Natural/ regulated	Lake Geneva's water level is regulated by the Seujet Dam in Geneva, which controls the outflow to the Rhône river. The dam maintains the average surface elevation at approximately 372 m.a.s.l. while it also manages the outflow to balance competing demands such as flood control and water supply. The overall management of the lake is a collaborative effort between Switzerland and France through the International Commission for the Protection of the Waters of Lake Geneva (CIPEL).
Distance from upstream station	6 km
Distance to downstream station	0 km, directly at the lake outlet
Hydrological regime	Nival regime (at both the up- and downstream station)
Lake Lucerne	
Lake surface area	114 km ²
Natural/ regulated	According to the weir regulations, water level fluctuations in Lake Lucerne are permitted as in an unregulated lake. The regulation ensures a largely natural water level regime within certain tolerance limits. Regulation is only applied to prevent the lake level from falling below 433.45 m.a.s.l. or reaching above 434 m.a.s.l. Average discharge at the lake outlet: 110 m ³ /s. Highest measured discharge: 473 m ³ /s.
Distance from upstream station	1.8 km
Distance to downstream station	1 km (just downstream of the weirs)
Hydrological regime	Nival regime (at both the up- and downstream station)
Lake Walen	
Lake surface area	24 km ²
Natural/ regulated	Natural (no lake level regulation)
Distance from upstream station	5 km
Distance to downstream station	0.8 km
Hydrological regime	Nival regime (at both the up- and downstream station)

Overall, I am not sure if this study deserves publication in the intended journal: the case study is certainly interesting but little specific new knowledge is presented.

Reply: The aim of this study is to improve the understanding of how river water temperature responds to atmospheric heatwaves. We believe that our study makes the following valuable and novel contributions to the field: (1) The large-sample perspective on water temperature and its extreme events (275 catchments) is unique and provides novel insights on the spatial variability of the relationship between atmospheric and riverine heatwaves over a heterogeneous and climate sensitive area like the European Alps; (2) the behavior of river water temperature during heatwaves has received little attention until now. Here, we for this first time show how different hydro-climatic conditions can strengthen or weaken the relationship between air temperature and river water temperature, as well as the development of riverine heatwaves; (3) Although previous studies have suggested the impact of meltwater on water temperature, we provide large-scale, data-based evidence for this hypothesis; (4) By shedding light on the interlinkages between water temperature and its hydro-climatic drivers, and by adopting a network perspective in the case studies, this study is

able to bring together many different disciplines, making it well-suited to the interdisciplinary audience of Communications Earth & Environment.

Detailed comments

Comment: How does your definition of heatwave compare to the one of other authors? Given that your definition allows heatwaves to occur in winter, could this not be misleading (terminology-wise) in comparison to previous work?

Reply: The use of the selected definition is very common for studies on riverine heatwaves and other types of aquatic heatwaves (e.g. lake or marine heatwaves (Hobday et al., 2016; Woolway et al., 2021)). From the perspective of river ecosystems, it is important to consider heatwaves that can occur year-round, because riverine heatwaves can be equally relevant to aquatic flora and fauna during both cold and warm months (e.g. for the reproduction of fish species (Jensen et al., 2008; Butzge et al., 2021)).

Atmospheric heatwaves are also mostly defined as periods of abnormally warm weather that last for multiple days, although there exists no single universal definition. The use of a 90th percentile varying threshold is quite common, but it is correct that many studies focus mainly on summer atmospheric heatwaves. Especially in combination with high humidity, these summer events have shown to have a high impact on human society (Perkins and Alexander, 2013). However, from a hydrological perspective, winter heatwaves in mountainous regions can also be highly significant, as they can lead to rapid snowmelt and reduced snow cover. To address your comment in our manuscript and to avoid confusion among readers, we have extended the methods section on 'Definition of heatwaves' as follows (Line 360-374):

"A heatwave occurs when temperatures show strong positive anomalies for several consecutive days. In line with existing definitions of aquatic and atmospheric heatwaves, we have adopted the following definition for atmospheric and riverine heatwaves in this study: A heatwave is a period during which the daily mean temperature exceeds a local, seasonally varying 90th percentile threshold for a minimum of five consecutive days. The threshold is defined as the 90th percentile of daily mean temperature, centered on a 30-day window and averaged over the 10-year reference period 2011--2021. This definition has been widely used in recent studies to identify riverine heatwaves, including those by Tassone et al. (2022a, 2022b), Sun et al. (2024, 2025), and Sadayappan et al. (2025). Regarding atmospheric heatwaves, the literature does not provide one single universal definition: the use of a 90th percentile threshold is common, but some studies focus only on summer heatwaves, while others include humidity to better capture the impact of atmospheric heatwaves on human health (Perkins and Alexander, 2013). However, winter atmospheric heatwaves can have a significant impact on mountain hydrology by triggering rapid snowmelt events. Similarly, from the perspective of river ecosystems, riverine heatwaves can be equally relevant to aquatic flora and fauna during both cold and warm months (e.g. for the reproduction of fish species (Jensen et al., 2008; Butzge et al., 2021)). Finally, for the purpose of this study, it is important that both atmospheric and riverine heatwaves are defined in identical ways to facilitate the comparison between the two phenomena, which is why we have selected the above-described definition for both heatwave types."

- Hobday, A. J., Alexander, L. V., Perkins, S. E., Smale, D. A., Straub, S. C., Oliver, E. C. J., Benthuyzen, J. A., Burrows, M. T., Donat, M. G., Feng, M., Holbrook, N. J., Moore, P. J., Scannell, H. A., Sen Gupta, A., and Wernberg, T. (2016). *A hierarchical approach to defining marine heatwaves*. *Progress in Oceanography*, 141:227–238.
- Woolway, R. I., Jennings, E., Shatwell, T., Golub, M., Pierson, D. C., and Maberly, S. C. (2021). *Lake heatwaves under climate change*. *Nature*, 589(7842):402–407
- Tassone, S. J., Besterman, A. F., Buelo, C. D., Ha, D. T., Walter, J. A., and Pace, M. L. (2022a). Increasing heatwave frequency in streams and rivers of the United States. *Limnology and Oceanography Letters*, 8(2):295–304.
- Tassone, S. J., Besterman, A. F., Buelo, C. D., Walter, J. A., and Pace, M. L. (2022b). Co-occurrence of Aquatic Heatwaves with Atmospheric Heatwaves, Low Dissolved Oxygen, and Low pH Events in Estuarine Ecosystems. *Estuaries and Coasts*, 45(3):707–720
- Sun, J., Di Nunno, F., Sojka, M., Ptak, M., Zhou, Q., Luo, Y., Zhu, S., and Granata, F. (2024). Long-term daily water temperatures unveil escalating water warming and intensifying heatwaves in the Odra river Basin, Central Europe. *Geoscience Frontiers*, 15(6):101916.
- Sun, J., Di Nunno, F., Sojka, M., Graf, R., Wrzesiński, D., Ptak, M., Dong, W., Xu, J., Zhou, Q., Luo, Y., Zhi, W., Noori, R., Zhu, S., and Granata, F. (2025). River Thermal Dynamics and Heatwaves of Polish Rivers Under Climate Change. *Water Resources Research*, 61(5):e2024WR039331.
- Sadayappan, K. and Li, L. (2025). Riverine heat waves on the rise, outpacing air heat waves. *Proceedings of the National Academy of Sciences*, 122(39):e2503160122. Publisher: Proceedings of the National Academy of Sciences.
- Perkins, S.E., and Alexander, L.V. (2013). On the measurement of heat waves. *Journal of Climate*, 26(13):4500-4517

Comment: Is there a difference between “availability of meltwater » and the « meltwater fraction of the runoff that potentially reaches the river » (see lines 139 and 140)

Reply: Yes, there is a significant difference: “availability of meltwater” is a binary descriptor for whether or not a catchment produces meltwater because of temperature-induced melting of available glaciers or snow cover. The “meltwater fraction of the runoff that potentially reaches the river” provides additional insights on how significant the meltwater inflow to the river is in comparison to other discharge sources such as rainfall runoff. For the cooling effect on the river, the meltwater fraction is more informative than the meltwater availability. To clarify the difference between the two concepts, we have reformulated these sentences (see Line 148-152):

“The mitigating effect of positive discharge anomalies in spring and summer seems linked to ~~the availability of meltwater and the meltwater fraction of the runoff that potentially reaches the river~~ that can potentially reach the river, which refers to the amount of meltwater in relation to the sum of meltwater, precipitation, and evaporation. In catchments with snow and ice, substantial meltwater inputs to the river can occur during the melting season in spring and, at higher elevations, also in summer (Figure 4c and d). [...]”

In addition, the definition of the meltwater fraction is also provided in the methods section (at line 353): “In our study, we use the catchment mean daily meltwater (mm d^{-1}), defined as the sum of snowmelt and glacier melt, and the meltwater fraction (-), defined as meltwater / (meltwater + precipitation - evaporation).”

Comment: Is the meteorological data set useful for the smaller catchments in areas with high topography? What percentage of the catchments are headwater catchments or catchments smaller e.g. than 300 km², for which the resolution of the meteo data is not good enough? Same question for the gridded daily snowmelt and glacier: what is the model resolution at the study latitude? And is it reliable for small catchments?

Reply: As mentioned in the Methods section, we obtained the hydro-climatic variables from the gridded Copernicus European Regional Re-Analysis (CERRA) dataset, which has a high spatial

resolution of 5.5 km. For the gridded daily snowmelt and glacier melt, we used the output from an updated version of the gridded global hydrological model PCR-GLOBWB, which has been specifically set up and evaluated for the Alps, which are characterized by complex topography. This model provides snowmelt and glacier melt data at a 30 arcsec resolution (1.1 km at the equator, which equals approximately 700 m latitude and 1.1 km longitude in the Alpine region). Both products provide data at a relatively high spatial resolution, which is indeed particularly valuable for the complex topography of the Alpine region with a high spatial variability of meteorological variables like temperature and precipitation.

Only a few catchments in our study (4%, n=12) have an area slightly smaller than the grid size of the CERRA dataset. Yet, the use of the selected meteorological data set is substantiated: CERRA has been shown to outperform lower-resolution products like ERA5 and ERA5-Land and has proven to be a preferred option compared to other large-scale climate reanalysis datasets (Wood et al., 2025). Finally, we think that using a single, large-scale dataset covering the entire Alpine region is preferable to using multiple, smaller-scale national datasets: while such national products may be of slightly higher quality and spatial resolution than the European-wide products, they often provide a smaller set of variables, and combining different products can introduce additional uncertainties.

We have added the following sentence to the methods section of our manuscript to justify our choice (Line 336):

“CERRA has been shown to outperform lower-resolution reanalysis products such as ERA5 and ERA5-Land (Wood et al., 2025). In addition, it provides homogeneous data across country borders.”

- Wood, R. R., Janzing, J., van Hamel, A., Götte, J., Schumacher, D. L., and Brunner, M. I. (2025). Comparison of high-resolution climate reanalysis datasets for hydro-climatic impact studies, *Hydrol. Earth Syst. Sci.*, 29, 4153–4178, <https://doi.org/10.5194/hess-29-4153-2025>.

Comment: The z-scoring procedure for precipitation and snowmelt is clear (line 370) but it is unclear how positive or negative z-scores during heatwaves can be interpreted in this case since the probability of precip being zero during the heatwaves is not part of that metric

Reply: Thank you for bringing this ambiguity to our attention. While checking the Methods section, we realized that this paragraph is actually outdated and needs to be removed. Initially, we analyzed meltwater and precipitation as separate variables, and this additional step was a valuable addition to dealing with the many zero values throughout the year. However, we later decided to focus on the meltwater fraction and water availability (in the form of precipitation minus evaporation) instead because these variables proved to be more informative for our analysis. Since these ‘new’ variables result in fewer zeros and have a clear annual cycle, it became unnecessary to apply this extra step.

Gallice, A., Schaefli, B., Lehning, M., Parlange, M. B., and Huwald, H.: Stream temperature prediction in ungauged basins: review of recent approaches and description of a new physics-derived statistical model, *Hydrol. Earth Syst. Sci.*, 19, 3727–3753, 10.5194/hess-19-3727-2015, 2015.

Michel, A., Brauchli, T., Lehning, M., Schaefli, B., and Huwald, H.: Stream temperature and discharge evolution in Switzerland over the last 50 years: annual and seasonal behaviour, *Hydrology and Earth System Sciences*, 24, 115–142, 10.5194/hess-24-115-2020, 2020.

Michel, A., Schaefli, B., Wever, N., Zekollari, H., Lehning, M., and Huwald, H.: Future water

temperature of rivers in Switzerland under climate change investigated with physics-based models, *Hydrol. Earth Syst. Sci.*, 2021, 1-45, 10.5194/hess-26-1063-2022, 2022.

Piccolroaz, S., Calamita, E., Majone, B., Gallice, A., Siviglia, A., and Toffolon, M.: Prediction of river water temperature: a comparison between a new family of hybrid models and statistical approaches, *Hydrological Processes*, 30, 3901– 3917, 10.1002/hyp.10913, 2016.

Piccolroaz, S., Toffolon, M., Robinson, C. T., and Siviglia, A.: Exploring and Quantifying River Thermal Response to Heatwaves, *Water*, 10, 1098, 2018.

Toffolon, M., and Piccolroaz, S.: A hybrid model for river water temperature as a function of air temperature and discharge, *Environmental Research Letters*, 10, 114011, 2015.

Reviewer #1 provided additional comments and textual suggestions to the pdf of the manuscript.

Some of the comments in the PDF have already been addressed above, so we won't discuss them again here. Below, we provide a short overview of the remaining comments, including how we addressed them:

Introduction:

L16: The use of the word 'Sudden'. Replaced by 'Substantial'. (see L18)

L19: Is there an example of fish die-offs in Europe/Alps? An example from 2022 was added. (L22)

L33: Suggestion to expand on water management examples by mentioning hydropower production and thermopeaking. We adopted this suggestion. (L35)

L35: Suggestion to expand 'groundwater inflow *and* outflow'. We adopted this suggestion. (L37)

Discussion:

L194: Is there a figure to show that the coupling between air and water temperature is elevation and season-dependent? Yes, this becomes visible from Figure 2b. We have clarified this in the text. (L206)

L199: "decoupling" is perhaps a bit a strong word here, what indicates decoupling? A similar comment was raised by reviewer 2. Therefore, the word 'decoupling' has been replaced throughout the manuscript, and this particular sentence has been rephrased as follows: "For these events, ~~air and water temperatures were decoupled~~ the thermal sensitivity of water to air temperature is weak, preventing the development of riverine heatwaves." (L210)

L209: 'seasonal precipitation surplus'; This is not studied, right? Do you mean precipitation – ET? We have rephrased the sentence as follows: 'Positive discharge anomalies are typically associated with a seasonal precipitation surplus or melting events.' In the remainder of the paragraph, we discuss how this relates to our observations. (L221)

L217: 'rivers fed by rainfall or thermally stable groundwater'; groundwater is not thermally stable (it still shows seasonality) but is simply warmer than the meltwater. We have rephrased this to "rivers being fed by warmer rainfall or groundwater" (L231). Later in that paragraph we explain the phenomenon in more detail (L235-240): "This rainfall can result in groundwater recharge and activate groundwater flow to the river. The inflow can provide the river with relatively warm water in winter (Tague et al., 2007; Hare et al., 2021), because groundwater temperature is known to be more

stable throughout the year. This could lead to anomalously high river water temperatures and the development of riverine heatwaves.”

L272: General comment on section 3.2 ‘Implications for water temperature response in a warming world’; As far as I see, this section does not capitalize results of this study but essentially published work. Yes, this is correct and intended. Given the broad readership of *Communications Earth & Environment* and the fact that not all readers will be familiar with the field of water temperature and riverine heatwaves, we believe it is important to provide context for our study and outline how our results can be interpreted in relation to existing studies that have examined future developments.

Reviewer 2:

Review of the manuscript:

This manuscript uses high-frequency observational datasets to investigate the response of riverine heatwaves to atmospheric heatwaves under different hydro-climatic conditions. The authors report that only 47% of atmospheric heatwaves lead to riverine heatwaves, and that the occurrence of riverine heatwaves often requires not only atmospheric forcing but also additional hydro-climatic conditions such as anomalous discharge and meltwater. Furthermore, the study presents a detailed case analysis of a river network to explore the interactions among riverine heatwaves, atmospheric heatwaves, and hydro-climatic variables.

Compared with previous studies, which primarily focused on long-term changes in riverine heatwave characteristics such as intensity, frequency and duration, then tended to attribute them in a simplified manner to atmospheric heatwaves, rising air temperatures, or discharge anomalies, this study provides a more systematic perspective. Specifically, it analyzes the hydro-climatic conditions under which atmospheric heatwaves are likely to trigger riverine heatwaves, thereby offering valuable insights into the driving mechanisms of riverine heatwaves.

While the manuscript addresses an important and timely topic, several aspects require clarification and further development:

Comment: The central question for me is how the authors define that riverine heatwaves are triggered by atmospheric heatwaves. Is this based solely on temporal overlap? If so, I am not convinced this is a robust approach, this only reflects coincidence or probability rather than causality. Recent studies (Wang et al., 2024a, b; Sun et al., 2025) have used model simulations and controlled experiments to explicitly determine the role of atmospheric heatwaves in driving freshwater heatwaves. The authors should clarify their criteria and consider whether this approach is robust, because it is the base and core of this manuscript.

Reply: In our study, we indeed analyze the response of water temperature to atmospheric heatwaves based on temporal overlap and use co-occurrence as an indication for association. There are two main reasons why we believe the chosen method is valid and robust and can provide valuable insights into the relationship between atmospheric and riverine heatwaves:

First, based on physical principles, river water temperature can be influenced by air temperature, while it is unlikely that air temperature is influenced by river water temperature. Given the general consensus in the literature that air temperature has a strong influence on water temperature, it is likely that the co-occurrence of increased water temperature with increased air temperature results from this relationship. However, if water temperature is hardly increasing (or even shows negative anomalies) despite the presence of an atmospheric heatwave, we need to consider a broader set of other hydro-climatic variables to explain this behaviour. Deviations in the anomalies of these variables serve as indicators of processes that may reduce the thermal sensitivity of water to air temperature.

Second, we deliberately chose to conduct an observation-based study, using water temperature observations rather than a model-based approach with reconstructed water temperature data. This is a valuable aspect of our study, which was explicitly recognized as such by Reviewer 3. Water

temperature models, such as the MARX and air2stream models (Sun et al., 2024), often use air temperature as their main input variable and therefore might not capture all the relevant processes that can influence water temperature under extreme conditions (van Hamel and Brunner, 2024). Aside from the high data requirements for setting up, calibrating and validating such models, the results also depend heavily on model accuracy, which is often only evaluated in terms of average performance and not specifically in terms of reconstructing extreme values. Large-scale observation-based studies, like ours, are scarce but necessary to complement existing model-based studies because they overcome these model limitations. However, observation-based studies often rely on association metrics to analyze data, identify patterns, test hypotheses, and draw conclusions. In our study, we applied a sample split to test the hypothesis that certain hydro-climatic conditions enhance the development of riverine heatwaves during atmospheric heatwaves. We created two sub-samples of reasonable size by differentiating between atmospheric heatwaves that co-occurred with/without riverine heatwaves. We showed that there is indeed a significant difference between the hydro-climatic conditions of the two event samples (statistical significance is proved by using the Mann-Whitney U test), which gives us sufficient confidence to draw conclusions based on this comparison.

Based on the above, we believe that our observational study is a valuable addition to the existing literature and that analyzing co-occurrence is a robust way to gain new insights into the relationship between different variables.

Comment: Line 87-88: The manuscript reports that the link between air and water temperatures is strongest in catchments at low elevations with a small snow fraction. Although Figure 1c seems to suggest this pattern, the figure needs added relationship value and statistical significance to support the conclusion.

Reply: To support our conclusion, we have added information on the Spearman correlation coefficient and its statistical significance to Figure 1c. The Spearman correlation coefficient ($= -0.668$) shows that the two variables, namely the percentage of precipitation falling as snow and the percentage of atmospheric heatwaves that result in riverine heatwaves, show a relatively strong (≥ 0.6) and negative monotonic relationship. In addition, both a linear regression model and a spline with 2 degrees of freedom highlight the significant relationship between the percentage of precipitation falling as snow and the percentage of atmospheric heatwaves that result in riverine heatwaves (p -value < 0.05), whereby the spline slightly outperforms OLS linear regression.

We have updated Figure 1c and its caption as given below:

Figure 1. (c) Relationship between the percentage of precipitation falling as snow per catchment and the percentage of atmospheric heatwaves resulting in riverine heatwaves (Spearman correlation coefficient = -0,668). Linear regression (orange) and a spline with 2 degrees of freedom (green) both highlight the significant relationship between the two variables (p -value < 0.05), whereby the spline slightly outperforms linear regression.

Comment: Line 110: The suitability of the term “decoupling” is questionable. Previous studies (e.g., Cassie., 2006) have shown that air and water temperatures follow an S-shaped relationship: linear in the mid-range but nonlinear at low and high extremes, where river water temperature ceases to increase proportionally with air temperature. Since riverine heatwaves are extreme events, it is unlikely that these processes represent a true “decoupling” between atmospheric and riverine heatwaves. The authors should reconsider their terminology and provide a more accurate interpretation.

Reply: 'Decoupling' is indeed a rather strong term, and we agree that it doesn't accurately reflect the situation. Instead of 'decoupling' we decided to describe more precisely what we observe, namely that 'the thermal sensitivity of water to air temperature is becoming weaker'. We have rephrased the three sentences, in which we used the term 'decoupling':

Line 116-120: “Similar WT and AT z-scores suggest a **strong coupling** between air and water temperature, whereby water temperature experiences similarly high positive anomalies as air temperature. This could *result* in the development of a riverine heatwave. Conversely, a significant difference between the AT and WT z-scores indicates a ~~decoupling between the two variables~~ *reduction in the thermal sensitivity of water to air temperature* because the water temperature is barely affected by the high air temperature anomalies.”

Line 210-212: “While half of the atmospheric heatwaves led to riverine heatwaves, the other half did not (Figure 1b). For these events, air and water temperatures were ~~decoupled~~ *the thermal sensitivity of water to air temperature is weak*, preventing the development of riverine heatwaves.”

Formerly Line 218-219: Such ~~decoupling~~ *is favoured by certain hydro-climatic conditions.* (Note: this sentence has been removed entirely)

Comment: Line 127: The manuscript emphasizes cases where atmospheric heatwaves coincide with riverine heatwaves. However, it would also be valuable to examine cases where riverine heatwaves

occur in the absence of atmospheric heatwaves. Do such cases exist? If so, comparing these with the co-occurrence cases could provide important insights into additional hydro-meteorological influences.

Reply: Yes, cases in which riverine heatwaves occur in the absence of atmospheric heatwaves do exist. As we state at Line 292: “Finally, 69% of the riverine heatwaves in our study could not be directly linked to an atmospheric heatwave”. We discuss this phenomenon in relation to existing literature, but we conclude that further research is needed to shed light on the processes that influence the development of these riverine heatwaves. These questions lie outside the scope of this study, since our focus is specifically on what happens during atmospheric heatwaves.

Comment: Line 154: The described phenomenon is puzzling. In general, solar radiation is an important driver of river water temperature, while high humidity, which as authors described often associated with rainfall, reduces solar radiation reaching the water surface and thereby suppresses warming. In the discussion, the authors suggest that high humidity and reduced radiation accompany warmer temperatures, rainy, or stormy weather, which may lead to riverine heatwaves. However, it is unclear whether rainy or stormy conditions trigger or suppress riverine heatwaves, and I am not aware of references supporting this interpretation. A more detailed analysis, supported by relevant literature, is needed to provide a convincing explanation of this mechanism.

Reply: Thank you for your comment. We agree that our observation may seem surprising at first: atmospheric heatwaves in winter and autumn which co-occur with slightly lower radiation and higher humidity have the tendency to more likely co-occur with riverine heatwaves. In the ‘Discussion’ section, we discuss how the observed positive discharge anomalies in autumn and winter could favour the development of riverine heatwaves. However, we haven't made it clear enough how we explain our observations in relation to humidity and radiation. We have extended our elaboration in the ‘Discussion’ section to provide a potential explanation for our observations (Line 229-240):

“In autumn and winter, an opposite signal is visible, with positive discharge anomalies favouring the development of riverine heatwaves (Figure 4a). At this time of the year, positive discharge anomalies are less *likely to be* related to meltwater, *which is mostly available in spring*, and instead indicate wet conditions, with rivers being fed by *warmer* rainfall or groundwater. *Our finding that riverine heatwaves in these seasons occur more frequently when atmospheric heatwaves are accompanied by lower radiation but higher humidity anomalies (Supplementary Figure 4c-d, k-l) also supports the hypothesis that rainfall plays a role in the development of positive discharge anomalies. Lower radiation and higher humidity could point towards cloudy and rainy weather, with warm air advection from the south-west entering the region under the influence of low pressure systems (Beniston, 2005; Holmberg et al., 2023). This rainfall can result in groundwater recharge and activate groundwater flow to the river. The inflow can provide the river with relatively warm water in winter (Tague et al., 2007; Hare et al., 2021), because groundwater temperature is known to be more stable throughout the year. This could lead to anomalously high river water temperatures and the development of riverine heatwaves.*”

- Beniston, M., Warm winter spells in the Swiss Alps: Strong heat waves in a cold season? A study focusing on climate observations at the Saentis high mountain site. *Geophysical Research Letters*, 32(1), 10.1029/2004GL021478 (2005).

- Holmberg, E., Messori, G., Caballero, R. & Faranda, D. The link between European warm-temperature extremes and atmospheric persistence. *Earth System Dynamics* 14, 737–765 (2023).
- Tague, C., Farrell, M., Grant, G., Lewis, S. & Rey, S. Hydrogeologic controls on summer stream temperatures in the McKenzie River basin, Oregon. *Hydrological Processes* 21, 3288–3300 (2007).
- Hare, D. K., Helton, A. M., Johnson, Z. C., Lane, J. W. & Briggs, M. A. Continental-scale analysis of shallow and deep groundwater contributions to streams. *Nature Communications* 12, 1450 (2021).

Comment: Line 159: Why were these five stations in Drau River chosen? Were they the only locations where this phenomenon was observed, or were there other criteria guiding the selection? A more transparent justification would be helpful.

Reply: To select rivers and stations for the network perspective, we did apply a few criteria. To clarify our selection criteria, we have added this missing information to the Methods section (Line 424-427):

“This river and its stations were chosen for the following reasons: the river includes several stations that measure water temperature downstream of each other; the stations are not affected by any large lakes; human regulation is limited; and the catchments include high-elevation areas and a small glacier, which allows us to study the influence of meltwater on riverine heatwave development.”

Overall, while the study provides valuable insights, these clarifications are necessary before the manuscript can be considered for publication.

Reply: Thank you for acknowledging the value of our work and the constructive feedback provided.

Reference:

Wang, W. et al. The impact of extreme heat on lake warming in China. *Nat Commun* 15, 70 (2024).

Wang, X. et al. Disproportionate impact of atmospheric heat events on lake surface water temperature increases. *Nat. Clim. Chang.* (2024) doi:10.1038/s41558-024-02122-y.

Sun, J. et al. Impact of extreme atmospheric heat events on river thermal dynamics and heatwaves. *Journal of Hydrology* 659, 133292 (2025).

Caissie, D., 2006. The thermal regime of rivers: a review. *Freshw. Biol.* 51 (8), 1389-1406.

Reviewer 3:

Review of manuscript COMMSENV-25-4311 „River temperature response to atmospheric heatwaves is modulated by discharge and meltwater”

I. Summary of the most important scientific findings

(1) This paper addresses a very important problem of analyzing the potential for mitigation of atmospheric heatwaves in river systems due to additional meltwater distribution. In particular, river runoff regimes will be highly affected under climate change conditions, while in parallel snow lines are moving to larger altitudes and glaciers are significantly retreating.

(2) For a very large data set in Austrian Alps including longer term data (2011-2021) from 275 catchments in Austria and Switzerland, atmospheric heat waves as well as riverine heat waves were identified and analyzed.

(3) Their results demonstrated that the river temperature response can be very different, depending on the location of the catchment/temperature measurement. Only for 47% of the heatwaves analyzed lead to riverine heatwaves.

(4) This was controlled to a large extent by discharge level and melt water contributions, with high discharge and high meltwater contributions preventing riverine heatwaves, while a strong link exists between low discharge and no-meltwater contributions and the generation of riverine heat waves.

II. Novelty and contributions of this paper

(5) While such relationships/effects between air temperature and river temperature (or atmospheric and riverine heatwaves) have been found investigated previously (but only using smaller number of catchments and/or using modelling results), I believe this study is of very high value to the scientific community as it is based on a large number of locations in a very heterogeneous and climate sensitive area in Central Europe. In particular, I like that this study is based measured/observed discharge & temperature data.

(6) Also, making these available to larger scientific community is a very valuable contribution to a large number of research institutes

III. Critical comments and recommendations to the editor

(7) I have very much enjoyed reading this manuscript. It is highly relevant to hydrologist with some focus on climate change related topics in water resource management or ecology.

However, there are two may be more critical questions or concerns I have that are listed in the following:

Reply: Thank you very much for highlighting the value and relevance of our work and for providing constructive feedback on how to strengthen our manuscript.

Comment:

(8) How did you deal with the time lag between temperature induced melting and runoff routing to the measurement site – maybe I missed such a statement.

Reply: The time lag between induced melting, runoff and river discharge is related to the catchment area, main channel length, slope, soil type, etc. In our study, we did not explicitly account for a catchment specific time lag because we are working with catchments with a rather short response

time and we use a daily time resolution. We assume that the time lag in most headwater catchments is rather small (< 1 day), since 76 % of the catchments part of our dataset have a catchment area smaller than 900 km² (30km² x 30km²). In larger catchments, the time lag is potentially longer than one day. However, since our analysis focuses on processes that take place during atmospheric heatwaves, which (by definition) have a duration of at least 5 days, we expect to capture relevant increases in meltwater within this time window (also for larger catchments).

We have added the following sentence to the Methods section (Line 355-358):

“We do not explicitly account for catchment-specific time lags between temperature-induced melting and runoff routing, since these lags are assumed to be very short (shorter than one day for most headwater catchments and within five days for larger catchments, which lies within the minimum duration of an atmospheric heatwave).”

Comment: (9) You used PCR-GLOBEWB as a hydrological model that to my knowledge uses a Day-Degree method for modelling melting processes. We see in our analysis that there is a significant difference to more complex radiation driven approaches especially during melting – can you comment on how this would affect the analysis in your study. I guess commenting on this in the method section would be needed.

Reply: Snow and meltwater are derived from the PCR-GLOBWB2.0 model, which indeed uses a temperature-index snow model.

Temperature-index models are commonly used in literature because of their low computational costs, their ability to run on widely available input data, and their generally good model accuracy (Hock, 2003). At the same time, we acknowledge that temperature-index models do have shortcomings in representing detailed spatial variability and short-term temporal variability in snow melt rates. Physically based approaches, such as energy balance models, provide a better representation of such detailed snow processes (Magnussen et al., 2015). However, such models require much more detailed input data and are not always available on a large scale (e.g. Alpine 3D is generally applied at smaller scales). Furthermore, there are also studies that show that errors in the input and validation data, rather than the model structure, have the greatest effect on model performance (Magnussen et al., 2015). Therefore, we argue that calibration and validation of the temperature-index snow model against a detailed snow reanalysis product (as done in the model framework chosen for our analyses) should give realistic estimates of melt contributions, especially when interested in daily discharge at a catchment scale. In addition, our study requires a consistent product across the entire Alpine domain, which is ensured by using a large-scale model such as PCR-GLOBWB. In contrast, we could not have guaranteed such consistency when using country-specific physically based products, because these products have been derived based on different modeling approaches, resolutions, and assumptions. Such inconsistencies would be suboptimal for our study because we work with some catchments that overlap multiple countries' boundaries, which would cause issues when using country-bound products.

Nevertheless, we acknowledge that temperature-index based models can have a lower accuracy during complex meteorological conditions (e.g. rain-on-snow events) compared to more detailed and elaborate physics-based models (Mott et al., 2023). To make readers aware of this potential

limitation, we added a sentence on the potential effect of using a temperature-index instead of a physically-based snow model to our study (Line 341-353):

“Gridded daily snowmelt (mm d^{-1}) and glacier melt (mm d^{-1}) were simulated at a 30 arcsec (approx. 1 km at the equator) resolution with the gridded global hydrological model PCR-GLOBWB 2.0 (Sutanudjaja et al., 2018; Hoch et al., 2023) for the period 2011-2021 and together they present the total daily melt. Specifically, we used an updated version of this model that has been set up and evaluated for the Alps by Janzing et al. (2025) and contains an updated snowmelt routine (with an expanded temperature index model and regional calibration) and a new glacier routine. *To ensure spatial consistency in snow data across country borders, we used a large-scale snow product generated using a model with a temperature-index based snow routine that has been thoroughly evaluated over the Alpine domain (Janzing et al., 2025), instead of country-specific products that rely on more detailed process representation. As input, the PCR-GLOBWB 2.0 model uses gridded precipitation and downscaled air temperature data. To obtain the downscaled air temperature product from CERRA, we followed the downscaling approach of Van Jaarsveld et al. (2025). We calculated correction factors by comparing monthly climatologies (1990-2019) for the CERRA temperature with the climatologies of the high-resolution Climatologies at High resolution for the Earth’s Land Surface Areas (CHELSA) V2.1 data set (Karger et al., 2017; Karger et al., 2021).*”

- Hock, R. (2003). Temperature index melt modelling in mountain areas, *Journal of Hydrology*, 282(1-4), 104-115, [https://doi.org/10.1016/S0022-1694\(03\)00257-9](https://doi.org/10.1016/S0022-1694(03)00257-9)
- Girons Lopez, M., Vis, M. J. P., Jenicek, M., Griessinger, N., and Seibert, J. (2020). Assessing the degree of detail of temperature-based snow routines for runoff modelling in mountainous areas in central Europe, *Hydrology and Earth System Sciences*, 24, 4441–4461, <https://doi.org/10.5194/hess-24-4441-2020>.
- Magnusson, J., Wever, N., Essery, R., Helbig, N., Winstral, A., and Jonas, T. (2015). Evaluating snow models with varying process representations for hydrological applications, *Water Resources Research*, 51, 2707–2723, <https://doi.org/10.1002/2014WR016498>.
- Mott, R., Winstral, A., Cluzet, B., Helbig, N., Magnusson, J., Mazzotti, G., Quéno, L., Schirmer, M., Webster, C., and Jonas, T. (2023). Operational snow-hydrological modeling for Switzerland, *Frontiers in Earth Science*, 11, <https://doi.org/10.3389/feart.2023.1228158>.

I have a number of minor/specific comments that should be relatively easy to address, so that I would recommend to the editor to accept the manuscript after some revisions.

IV. Minor and specific comments

Comment: L15: I would already define heatwave here early in the introduction – you kept me asking this during the next lines until it appears in L64ff

Reply: Thank you for your comment, which we incorporated in the first paragraph (L15):

“River systems are increasingly exposed to atmospheric heatwaves, *which are defined as periods of at least 5 consecutive days during which the temperature exceeds a local, seasonally varying 90th percentile threshold (Perkins and Alexander, 2013).*”

Comment: L87: Define AT and WT.

Reply: We have incorporated this comment as follows (L97):

“Such coincidence results from a strong coupling between air and water temperatures as indicated by similarly high *five-day mean air temperature (AT) and water temperature (WT) z-scores.*”

Comment: L108: *potentially* sounds pretty vague!

Reply: We have adjusted the sentence by removing the word ‘potentially’ (L118):

“Similar WT and AT z-scores suggest a strong coupling between air and water temperature, whereby water temperature experiences similarly high positive anomalies as air temperature. This could *potentially* result in the development of a riverine heatwave.”

Comment: L154: I would be interested in why high humidity and rainy weather favor heat waves?

Reply: Thank you for highlighting this point. We agree that this is an interesting finding that requires further discussion. In the Discussion section, we already discussed how the observed positive discharge anomalies in autumn and winter could favour the development of riverine heatwaves. However, we did not establish a clear link with our findings relating to high humidity. Based on your comment, we have expanded this paragraph to provide a clearer explanation (Lines 229-240):

“In autumn and winter, an opposite signal is visible, with positive discharge anomalies favouring the development of riverine heatwaves (Figure 4a). At this time of the year, positive discharge anomalies are less *likely to be* related to meltwater, *which is mostly available in spring*, and instead indicate wet conditions, with rivers being fed by rainfall or thermally *more* stable groundwater. *Our finding that riverine heatwaves in these seasons occur more frequently when atmospheric heatwaves are accompanied by lower radiation but higher humidity anomalies (Supplementary Figure 4c-d, k-l) also supports the hypothesis that rainfall plays a role in the development of positive discharge anomalies. Lower radiation and higher humidity could point towards cloudy and rainy weather, with warm air advection from the south-west entering the region under the influence of low pressure systems (Beniston, 2005; Holmberg et al., 2023). This rainfall can result in groundwater recharge and activate groundwater flow to the river. The inflow can provide the river with relatively warm water in winter (Tague et al., 2007; Hare et al., 2021), because groundwater temperature is known to be more stable throughout the year. This could lead to anomalously high river water temperatures and the development of riverine heatwaves.*”

- Beniston, M., Warm winter spells in the Swiss Alps: Strong heat waves in a cold season? A study focusing on climate observations at the Saentis high mountain site. *Geophysical Research Letters*, 32(1), 10.1029/2004GL021478 (2005).

Comment: L240: I would argue that heat capacity of (lake) water is a physical constant and cannot be influenced

Reply: You are right, and our phrasing is indeed confusing. The total heat capacity of a lake cannot be reduced as a result of lake stratification, because it is an intrinsic property of the water. However, lake stratification does significantly impact how the lake stores and distributes heat. The layering reduces the vertical mixing of water, which traps heat in the upper layer and prevents it from reaching the deep water. We have rephrased the sentence as follows (Line 265-271):

“Seasonal differences in the thermal stability of lakes (i.e. the strength of thermal stratification, as discussed in North et al. (2013)) have been shown to significantly influence ~~the heat capacity~~ *the heat distribution* of lakes and the warming of downstream rivers: low thermal stability of lakes during the winter and early spring results in *vertical* mixing and ~~relatively high heat capacities~~ *a better distribution of heat*, while the high thermal stability of lakes in summer causes stratification, with a thin upper layer of water that is strongly influenced by atmospheric conditions (and thus stronger warming of the downstream river).”

Comment: L308: PCR-GLOBWB is a model that to my knowledge operates with a degree-day approach. Can you justify this compared to more complex snow models (I believe Alpine 3D is used in Switzerland?) An added statement in the section would be sufficient in my opinion!

Reply: For a more detailed response to your comment regarding the use of a temperature-index model instead of a physical model, please see our reply to your second comment (see above).

Dear Dr. Brunner,

Your manuscript titled "River temperature response to atmospheric heatwaves is modulated by discharge and meltwater" has now been seen by our reviewers, whose comments appear below. In light of their advice we are delighted to say that we are happy, in principle, to publish a suitably revised version in Communications Earth & Environment, provided you present a clear discussion of the limitations of your approach, including the limited representation of the influence of groundwater.

Reply: *Dear Dr. Colombo, thank you for highlighting the need to provide a clear discussion of the limitations of our approach and dataset used. We added a note highlighting that our study does not explicitly consider groundwater influences and a new section on 'limitations and future directions'.*

Reviewer #1:

I thank the authors for their answers and the detailed revisions, the manuscript has improved even if no complementary analysis was performed by the authors. I just would like to add here that I do not agree with the view that soil moisture or atmospheric water availability are proxies for groundwater droughts: hydrologic droughts are different from soil moisture or atmospheric droughts. In absence of groundwater data, an actual proxy for groundwater could e.g be derived from streamflow (e.g. based on a baseflow analysis). A final version should certainly be even more explicit on the fact that the influence of groundwater is not well represented in this analysis.

Reply: *Thank you for highlighting the need to explicitly mention that the influence of groundwater on riverine heatwaves is not explicitly represented in our analysis. We added the following text to the discussion section to clarify this: 'While a lack of groundwater contributions to streamflow may favour the development of riverine heatwaves, we did not explicitly consider groundwater influences here because of a lack of adequate observed groundwater data. In order to explicitly consider such influences, a proxy for groundwater could be derived from streamflow (e.g. based on a baseflow analysis) or using a hydrological model.' To avoid the impression that we are using soil moisture as a proxy for groundwater levels, we removed the sentences related to soil moisture from that same paragraph.*

Reviewer #2:

Review of the manuscript:

The authors have carefully responded to the comments raised in the first round of review, and the revisions and responses are generally clear and effective in addressing my previous concerns. The overall quality of the manuscript has improved substantially, and the revised version is clearly stronger than the original submission. I also appreciate

the authors' decision to rely on in situ observations, which represents a clear strength of this study. Given the limited number of studies that investigate riverine heatwaves based on observed river temperature data, this observational perspective provides an important and valuable contribution to the field.

The methodology adopted in the manuscript primarily identifies potential “triggering” mechanisms based on the temporal overlap between atmospheric and riverine heatwaves. This framework appears to focus more on describing compound or co-occurring heatwave events rather than establishing a clear causal relationship. In addition, the thermal response of river systems often exhibits pronounced lag effects due to hydrological and geomorphological controls, which may not be fully captured by simple temporal concurrence analyses. This point was also raised in my first-round review. At the same time, even if this approach may currently represent the best feasible methodology based on available observational datasets, its robustness and inherent limitations still warrant careful discussion. While the authors argue that this observation-based approach is preferable to previous modeling studies. I also acknowledged that numerical models, despite introducing their own sources of uncertainty, remain the most direct method for isolating atmospheric forcing through controlled, variable-specific experiments.

Therefore, I strongly recommend that the authors include a dedicated paragraph or section to introduce the limitations of the current methodology, as well as a balanced comparison of the strengths and weaknesses of this study approach relative to modeling-based methods used in previous studies.

Reply: *Thank you very much for acknowledging the value of our study and highlighting the need to discuss the limitations of the approach chosen in a more balanced way. We added the following ‘Limitations and future directions’ section to the discussion, which specifically highlights the potential value of hydrological models for future studies on riverine heatwaves and their influencing factors:*

‘Together, the large-sample analysis and the two case studies allowed us to gain new insights into the factors influencing riverine heatwave development based on observational data. While observations are considered the most reliable data source available, moving to the modeling world could complement the observation-based analyses performed in this study. Numerical water temperature models enable targeted modeling experiments and sensitivity tests to isolate and quantify the influence of specific atmospheric and land-surface conditions on riverine heatwave development. However, a clear isolation of individual physical effects also remains challenging in the modeling world because of different sources of uncertainty involved in the modeling process. Specifically, disentangling the influence of in-stream processes (e.g. streambed friction, turbulence, and river heat capacity) and local conditions around the stream remains challenging and it is yet unclear which of the two has a larger influence. Furthermore, it remains challenging to account for human influences, such as the effect of reservoir regulation, water withdrawals, and thermal effluents from industries, on the

relationship between air and water temperatures because of a lack of consistent information on regulation, particularly across the large, cross-border sample of catchments considered in our study. Such influences could in future studies be investigated using targeted modeling experiments.

Our findings demonstrate that a simplistic air-to-water temperature relationship is insufficient to capture river water temperature dynamics under extreme conditions. Therefore, we argue that, in order to model or predict water temperature along rivers, information beyond the two variables of absolute air temperature and discharge is required. Still, commonly used statistical water temperature models — which are an attractive, less computationally demanding alternative to process-based models — often rely solely on these two types of inputs (e.g. Van Vliet et al. 2023, Toffolon et al. 2015, and Michel et al. 2022). Understanding the spatial and temporal dynamics of water temperature and the development of riverine heatwaves requires novel scientific approaches that recognise the complex interactions of all mechanisms involved, potentially calling for the use of machine learning or hybrid modeling approaches (Zhu et al. 2020, Acuna-Espinoza et al. 2015).'

Reviewer #3:

The authors have addressed my comments very well, and I am more than happy to recommend the paper for publications.

I have also read through the other colleagues comments, and I also think the authors have done an excellent job in addressing them.

The overall manuscript has now become even stronger and I am looking forward to read it very soon in "Communications - Earth and Environment"

Reply: *Thank you very much for your reassessment and acknowledging the improvements made during revisions.*

River temperature response to atmospheric heatwaves is modulated by discharge and meltwater

We make use of the double anonymized peer review. The authors are known by the editor.

September 12, 2025

Abstract

Alpine rivers are becoming increasingly exposed to atmospheric heatwaves. Because of their strong relationship with air temperature, rivers can experience persistent heat anomalies, known as riverine heatwaves, which can have serious consequences for river ecosystems and economy. This study aims to improve our understanding of how river water temperature responds to atmospheric heatwaves by focusing on the interplay of various hydro-climatic variables that can strengthen or weaken the thermal sensitivity of rivers to such events. Our results show that the response of water temperature to atmospheric heatwaves can vary substantially, with only 47% of atmospheric heatwaves leading to riverine heatwaves. Riverine heatwave development can be prevented by positive anomalies in discharge and meltwater, while negative anomalies in discharge strengthen the link between atmospheric and river temperatures. Future changes in these hydro-climatic conditions will likely increase the sensitivity of Alpine rivers to atmospheric heatwaves.

Keywords: Riverine heatwaves, atmospheric heatwaves, river water temperature, snowmelt, discharge, river ecology, mountain rivers, European Alps

1 Introduction

River systems are increasingly exposed to atmospheric heatwaves. These heatwaves can have a significant impact on river water temperature because rivers are highly sensitive to changes in air temperature [1; 2; 3]. Sudden increases in water temperature can have serious consequences for river ecosystems [4; 5], society [6], and the economy (e.g. energy production [7; 8] and fisheries [9]), particularly if water temperature remains extremely high for an extended period of time during so called riverine heatwaves [10]. Examples of the negative impacts of riverine heatwaves include the massive fish die-offs in Alaska and in Japan in 2019 and 2021, respectively [11; 12], as well as the reduction in nuclear power production at the Beznau power plant in Switzerland in the summer of 2025 due to a lack of cooling water [13].

Riverine heatwaves are likely becoming more frequent because of the strong relationship between water and air temperature [10; 14; 15]. Numerous studies have demonstrated that river water temperature has increased both locally and globally due to global warming [16; 14; 17; 18]. In addition, the frequency, persistence, and severity of atmospheric heatwaves are increasing [19], which has led to an increase in the frequency, duration, and intensity of aquatic heatwaves including marine [20; 21], lake [22; 23], and riverine heatwaves [14; 15; 24; 10; 3].

Even though air temperature is a strong determinant of river water temperature in lowland rivers, it is not the only variable influencing river thermal dynamics [1; 25; 2]. In addition, river water temperature is modulated by other atmospheric variables (e.g. radiation, humidity), hydrologic conditions (the amount of discharge and the interaction with different water bodies, e.g. lakes and tributaries), processes at the water-streambed interface (e.g. groundwater-surface water interactions), the surrounding environment (e.g. riparian vegetation and shading), and water management (e.g. thermal effluents from industries and urban wastewater) [1; 26; 27; 2]. In small (headwater) rivers in particular, air temperature is not always an accurate indicator of river water temperature, because local factors such as groundwater inflow, soil moisture, snow water equivalent, impoundments, stream size, and shading by riparian vegetation can play an important role too [28; 1; 29; 30; 31]. Substantial groundwater or meltwater inflow can potentially mitigate the effect of high air temperatures on river warming [32; 33]. Therefore,

we hypothesize that the origin and temperature of the water feeding a stream are probably as relevant as the amount of discharge (and the river’s thermal capacity) in controlling river 
[revised manuscript text omitted]

including negative WT z-scores (Figure 2b), regardless of catchment elevation or season. However, a slightly
 larger spread, and thus more negative WT z-scores, is visible in spring and, for high-elevation rivers (> 2000 m),
 in summer.

Figure 2: **Seasonal response of water temperature z-scores during atmospheric heatwaves.** (a) Water temperature (WT) and air temperature (AT) five-day mean z-scores for all time steps and catchments (in grey) and the maximum five-day mean z-scores during atmospheric heatwaves (in red), with a zoom-in density plot on the right. (b) Spread in average five-day mean WT z-scores during atmospheric heatwaves, categorised by catchment mean elevation and season: December-February (DJF), March-May (MAM), June-August (JJA), September-November (SON). The boxes represent the interquartile range, with the line indicating the median. The whiskers extend to points up to 1.5 times the box range. Outliers are plotted as individual points beyond the whiskers.

[revised manuscript text omitted]

Finally, the development of riverine heatwaves seems to be favoured by atmospheric conditions characterized
 by slightly low anomalies in surface net solar radiation but high and positive anomalies in relative humidity
 (Supplementary Figure 4c-d, k-l). High anomalies in relative humidity play a particularly significant role in
 autumn and winter, as well as at higher elevations. In other words, riverine heatwaves occur more frequently in
 these seasons when there is a comparably low energy input (radiation) but high humidity (and potentially rainy
weather).
**2.3 Water temperature response to atmospheric heatwaves changes along a river** 157 **network**
To understand how the water temperature response to atmospheric heatwaves can vary along river networks, we
compared the water temperature response of five different stations along the Drau river (Austria) to two similarly
severe atmospheric heatwaves in June 2015 and 2017. The 2015 heatwave occurred between 3rd and 7th of June
and affected all five stations along the Drau river network, with similarly high AT z-scores recorded at all stations
(Figure 5a and e, in purple). During this heatwave, the entire river showed a significant warming, as highlighted
by the simultaneous increase in WT z-scores across all stations during this period (Figure 5f). However, the
river’s water temperature did not reach z-scores similar to those of air temperature, with the result that no
riverine heatwaves were observed at any of the stations (Figure 5b, in purple). Two years later in 2017, another
atmospheric heatwave was observed between 19th and 24th of June. This heatwave affected all stations, but was
only labelled as such at the three lowest stations (Figure 5a and i, in pink). The AT z-scores observed during this
heatwave were very similar to those observed during the June 2015 heatwave. However, unlike the 2015 event,
this atmospheric heatwave resulted in riverine heatwaves at all five stations (Figure 5b, pink squares).
To understand why the response of water temperature to these relatively similar atmospheric heatwaves is so
different, we need to consider the discharge and meltwater during these events. In June 2015, positive anomalies
in discharge (Qs) were observed at all stations during the atmospheric heatwave (see Figure 5c and g), whereas in
June 2017, discharge showed much lower positive anomalies or even negative anomalies at the most downstream
station (Figure 5c and k). A similar pattern is found for meltwater (Figure 5d, h, and l): positive anomalies were
observed at all stations during the 2015 event, whereas mostly negative anomalies were observed during the 2017
event. In other words, the event in 2015 was characterized by anomalously high meltwater and discharge, while
the 2017 event was characterized by low discharge and meltwater because of an already disappearing snowpack.
The effect of these differences in discharge and meltwater on water temperature is striking: in 2015, no riverine
heatwave developed at any station, while in 2017, riverine heatwaves developed at all stations, including those
that did not even experience an atmospheric heatwave.
To assess the influence of lakes on the water temperature response, we compare the WT z-scores upstream and
downstream of five big lakes (Figure 6) during the seven year-round most severe atmospheric heatwaves observed
in our study domain, i.e. those that affected more than 50% of the catchments in our study domain. This lake
effect depends on the season: in winter and spring (see dots), the WT z-scores downstream of the lakes are in
general lower than the z-scores upstream of the lakes. In other words, the river downstream of the lake has
warmed less (compared to average water temperature) than upstream of the lake. In mid-summer (triangles)
and late-summer (crosses), the WT z-scores tend to show an opposite sign with higher WT z-scores downstream
of the lake than upstream of the lake. Now, the warming of the river is stronger downstream of the lake than
upstream.
**3 Discussion**
**3.1 Discussion and conclusions**
Our results show that about half (46%) of the atmospheric heatwaves observed in the European Alps over the
period 2011 to 2021 led to riverine heatwaves, indicating a strong coupling between air and water temperatures
(Figure 1b). This coupling is elevation and season-dependent: it is strongest in spring and summer at low
elevations, while it is strongest in autumn at high elevations. A similar elevation-dependent coupling was observed
by [33], who documented high air-to-water temperature sensitivity in low-land rivers but complete disconnection
for high-elevation snow-fed and regulated rivers using 19 catchments in Switzerland.
While half of the atmospheric heatwaves led to riverine heatwaves, the other half did not (Figure 1b). For these
events, air and water temperatures were decoupled, preventing the development of riverine heatwaves. Such
decoupling is favoured by certain hydro-climatic conditions. In particular, positive anomalies in discharge and
meltwater, which showed a clear cooling effect on rivers (Figure 3 and 4e), similar to what was documented by

Figure 5: **Downstream tracking of the hydro-climatic response to two atmospheric heatwaves along five stations in the Drau river.** (a-d) The maximum z-scores per station (from the upstream station 1 to the most downstream station 5) are shown for the two events: June 2015 in purple and June 2017 in pink. When stations experience an atmospheric heatwave or riverine heatwave, they are labelled with a diamond or square, respectively. (e-l) Temporal variations in air temperature (AT), water temperature (WT), discharge (Qs) and meltwater (Melt) z-scores for five stations along the Drau River during the atmospheric heatwave of June 2015 (e-h), and June 2017 (i-l).

[revised manuscript text omitted]

We adjusted the approach slightly for variables that naturally contain many zero values throughout the year,
such as snowmelt and precipitation. For these variables, we split the time series between wet and dry days by
introducing a conditional probability distribution. In the case of precipitation, for example, we state that days
with >1 mm precipitation are wet days and we calculate the probability for wet days to occur; on average there
were 68.7% wet days per catchment (min = 43.1%, max = 83.7%). Next, we follow the same steps as described
above, i.e. we calculate the Weibull plotting position and perform the transformation to z-scores, but only based
on the non-zero values (the wet days only). The same approach is applied to snowmelt, for which we observe on
average 26.6% melt-days per catchment (min = 4.3%, max = 64.2%).
Finally, we checked for each atmospheric heatwave whether it overlapped with any riverine heatwave. Any
atmospheric heat wave that overlaps with a riverine heatwave, even for just a few days, is labelled as such,
regardless of whether the riverine heatwave starts or ends before or after the atmospheric heatwave. Next, we
compared the atmospheric heatwaves that co-occurred with riverine heatwaves with those that did not, in order to
identify the conditions that strengthen or weaken the development of a riverine heatwave during an atmospheric
heatwave.

4.4 Case studies with network perspective

We conducted two case studies to demonstrate the variations in river water temperature sensitivity to atmospheric heatwaves along river networks. The first case study compared the responses of water temperature of five different measurement stations along the Drau river (Supplementary Figure 5) during two similarly severe atmospheric heatwaves. This river has its origin at the glaciers of the Großvenediger mountain in the National Park 'Hohe Tauern' (South-eastern Austria). The highest measurement station is located at 1686 m.a.s.l. and the lowest at 555 m.a.s.l, therewith covering an elevation range of 1131 m over a vertical distance of approximately 100 km. The second case study focussed on the effect of large lakes on the water temperature response. Specifically, we compared the water temperature z-scores during atmospheric heatwaves directly upstream and downstream of five big lakes in Switzerland, namely Lake Geneva, Bielersee, Brienersee, Vierwald-Stättersee, and Walensee.

Data availability

Note: *We make use of the double anonymized peer review. The data will be made available through HydroShare, and the link to access the data is known by the editor.*

References

- [1] Caissie, D. The thermal regime of rivers: a review. *Freshwater Biology* **51**, 1389–1406 (2006).
- [2] Leach, J. A., Kelleher, C., Kurylyk, B. L., Moore, R. D. & Neilson, B. T. A primer on stream temperature processes. *WIREs Water* **10**, e1643 (2023).
- [3] Sun, J. *et al.* Impact of extreme atmospheric heat events on river thermal dynamics and heatwaves. *Journal of Hydrology* **659**, 133292 (2025).
- [4] Johnson, M. F. *et al.* Rising water temperature in rivers: Ecological impacts and future resilience. *WIREs Water* **11**, e1724 (2024).
- [5] Sabater, S. *et al.* Extreme weather events threaten biodiversity and functions of river ecosystems: evidence from a meta-analysis. *Biological Reviews* **98**, 450–461 (2023).
- [6] Delpla, I., Jung, A. V., Baures, E., Clement, M. & Thomas, O. Impacts of climate change on surface water quality in relation to drinking water production. *Environment International* **35**, 1225–1233 (2009).
- [7] van Vliet, M. T. H. *et al.* Vulnerability of US and European electricity supply to climate change. *Nature Climate Change* **2**, 676–681 (2012).
- [8] van Vliet, M. T. H., Wiberg, D., Leduc, S. & Riahi, K. Power-generation system vulnerability and adaptation to changes in climate and water resources. *Nature Climate Change* **6**, 375–380 (2016).
- [9] Barange, M. *et al.* *Impacts of climate change on fisheries and aquaculture: synthesis of current knowledge, adaptation and mitigation options* (Food and Agriculture Organization of the United Nations (FAO), Rome, 2018).
- [10] Tassone, S. J. *et al.* Increasing heatwave frequency in streams and rivers of the United States. *Limnology and Oceanography Letters* **8**, 295–304 (2022).
- [11] Biela, V. R. *et al.* Premature Mortality Observations among Alaska's Pacific Salmon During Record Heat and Drought in 2019. *Fisheries* **47**, 157–168 (2022).
- [12] Kuroda, M., Kuroki, M., Kurokawa, D., Takeda, K. & Morita, K. A heatwave-related mortality event of endangered Sakhalin taimen *Parahucho perryi* in northern Hokkaido, Japan. *Ichthyological Research* (2025).
- [13] swissinfo.ch, S. W. I. Swiss nuclear power plant shuts down reactor due to the heat (2025). Accessed on 31/7/2025 from <https://www.swissinfo.ch/eng/climate-adaptation/beznau-nuclear-power-plant-shuts-down-one-of-its-reactors-due-to-the-heat/89617209>.

- [14] Sun, J. *et al.* Long-term daily water temperatures unveil escalating water warming and intensifying heatwaves
in the Odra river Basin, Central Europe. *Geoscience Frontiers* **15**, 101916 (2024).
- [15] Zhou, Q. *et al.* Characteristics of river heatwaves in the Vistula River basin, Europe. *Heliyon* **10**, e35987
(2024).
- [16] van Vliet, M. T. H., Ludwig, F., Zwolsman, J. J. G., Weedon, G. P. & Kabat, P. Global river temperatures
and sensitivity to atmospheric warming and changes in river flow. *Water Resources Research* **47** (2011).
- [17] Seyedhashemi, H. *et al.* Regional, multi-decadal analysis on the Loire River basin reveals that stream
temperature increases faster than air temperature. *Hydrology and Earth System Sciences* **26**, 2583–2603
(2022).
- [18] Michel, A., Brauchli, T., Lehning, M., Schaepli, B. & Huwald, H. Stream temperature and discharge evolution
in Switzerland over the last 50 years: annual and seasonal behaviour. *Hydrology and Earth System Sciences*
**24**, 115–142 (2020).
- [19] Domeisen, D. I. V. *et al.* Prediction and projection of heatwaves. *Nature Reviews Earth & Environment* **4**,
36–50 (2023).
- [20] Holbrook, N. J. *et al.* A global assessment of marine heatwaves and their drivers. *Nature Communications*
**10**, 2624 (2019).
- [21] Oliver, E. C. J. *et al.* Longer and more frequent marine heatwaves over the past century. *Nature Communi-*
*cations* **9**, 1324 (2018).
- [22] Woolway, R. I., Anderson, E. J. & Albergel, C. Rapidly expanding lake heatwaves under climate change.
*Environmental Research Letters* **16**, 094013 (2021).
- [23] Wang, X., Shi, K., Qin, B., Zhang, Y. & Woolway, R. I. Disproportionate impact of atmospheric heat events
on lake surface water temperature increases. *Nature Climate Change* **14**, 1172–1177 (2024).
- [24] Zhu, S. *et al.* An optimized NARX-based model for predicting thermal dynamics and heatwaves in rivers.
*Science of The Total Environment* **926**, 171954 (2024).
- [25] Laizé, C. L. R., Bruna Meredith, C., Dunbar, M. J. & Hannah, D. M. Climate and basin drivers of seasonal
river water temperature dynamics. *Hydrology and Earth System Sciences* **21**, 3231–3247 (2017). Publisher:
Copernicus GmbH.
- [26] Webb, B. W., Hannah, D. M., Moore, R. D., Brown, L. E. & Nobilis, F. Recent advances in stream and
river temperature research. *Hydrological Processes* **22**, 902–918 (2008).
- [27] Ouellet, V. *et al.* River temperature research and practice: Recent challenges and emerging opportunities for
managing thermal habitat conditions in stream ecosystems. *Science of The Total Environment* **736**, 139679
(2020).
- [28] van Hamel, A. & Brunner, M. I. Trends and Drivers of Water Temperature Extremes in Mountain Rivers.
*Water Resources Research* **60**, e2024WR037518 (2024).
- [29] Beaufort, A., Diamond, J. S., Sauquet, E. & Moatar, F. Spatial extrapolation of stream thermal peaks using
heterogeneous time series at a national scale. *Hydrology and Earth System Sciences* **26**, 3477–3495 (2022).
- [30] Fellman, J. B. *et al.* Stream temperature response to variable glacier coverage in coastal watersheds of
Southeast Alaska. *Hydrological Processes* **28**, 2062–2073 (2014).
- [31] Moatar, F. & Gailhard, J. Water temperature behaviour in the River Loire since 1976 and 1881. *Comptes*
*Rendus Geoscience* **338**, 319–328 (2006).
- [32] Moore, R. D. *et al.* Glacier change in western North America: influences on hydrology, geomorphic hazards
and water quality. *Hydrological Processes* **23**, 42–61 (2009).
- [33] Piccolroaz, S., Toffolon, M., Robinson, C. T. & Siviglia, A. Exploring and Quantifying River Thermal
Response to Heatwaves. *Water* **10**, 1098 (2018).
- [34] Anderson, S. & Chartrand, S. A century of variability of heatwave-driven streamflow in melt-driven basins
and implications under climate change. *Environmental Research Letters* **19**, 114059 (2024).
- [35] Zappa, M. & Kan, C. Extreme heat and runoff extremes in the Swiss Alps. *Natural Hazards and Earth
System Sciences* **7**, 375–389 (2007).
- [36] Feng, M., Zolezzi, G. & Pusch, M. Effects of thermopeaking on the thermal response of alpine river systems
to heatwaves. *Science of The Total Environment* **612**, 1266–1275 (2018).
- [37] Sohrabi, M. M., Benjankar, R., Tonina, D., Wenger, S. J. & Isaak, D. J. Estimation of daily stream water
temperatures with a Bayesian regression approach. *Hydrological Processes* **31**, 1719–1733 (2017).
- [38] McGregor, G. Synoptic Scale Atmospheric Processes and Heatwaves. In McGregor, G. (ed.) *Heatwaves:
Causes, Consequences and Responses*, 207–259 (Springer International Publishing, Cham, 2024).
- [39] Fischer, E. M., Seneviratne, S. I., Lüthi, D. & Schär, C. Contribution of land-atmosphere coupling to recent
European summer heat waves. *Geophysical Research Letters* **34** (2007).
- [40] Mukherjee, S. & Mishra, A. K. Increase in Compound Drought and Heatwaves in a Warming World.
*Geophysical Research Letters* **48**, e2020GL090617 (2021).
- [41] Constantz, J. Interaction between stream temperature, streamflow, and groundwater exchanges in alpine
streams. *Water Resources Research* **34**, 1609–1615 (1998).
- [42] Tague, C., Farrell, M., Grant, G., Lewis, S. & Rey, S. Hydrogeologic controls on summer stream temperatures
in the McKenzie River basin, Oregon. *Hydrological Processes* **21**, 3288–3300 (2007).
- [43] Hare, D. K., Helton, A. M., Johnson, Z. C., Lane, J. W. & Briggs, M. A. Continental-scale analysis of
shallow and deep groundwater contributions to streams. *Nature Communications* **12**, 1450 (2021).
- [44] White, J. C. *et al.* Drought impacts on river water temperature: A process-based understanding from
temperate climates. *Hydrological Processes* **37**, e14958 (2023).
- [45] Bruno, G. *et al.* Disentangling the role of subsurface storage in the propagation of drought through the
hydrological cycle. *Advances in Water Resources* **169**, 104305 (2022).
- [46] van Vliet, M. T. H. *et al.* Global river water quality under climate change and hydroclimatic extremes.
*Nature Reviews Earth & Environment* **4**, 687–702 (2023).
- [47] Mosley, L. M. Drought impacts on the water quality of freshwater systems; review and integration. *Earth-
Science Reviews* **140**, 203–214 (2015).
- [48] Graham, D. J., Bierkens, M. F. P. & van Vliet, M. T. H. Impacts of droughts and heatwaves on river water
quality worldwide. *Journal of Hydrology* **629**, 130590 (2024).
- [49] North, R. P. *et al.* The physical impact of the late 1980s climate regime shift on Swiss rivers and lakes.
*Inland Waters* **3**, 341–350 (2013).
- [50] van Vliet, M. T. H. *et al.* Global river discharge and water temperature under climate change. *Global
Environmental Change* **23**, 450–464 (2013).
- [51] Toffolon, M. & Piccolroaz, S. A hybrid model for river water temperature as a function of air temperature
and discharge. *Environmental Research Letters* **10**, 114011 (2015). Publisher: IOP Publishing.
- [52] Michel, A. *et al.* Future water temperature of rivers in Switzerland under climate change investigated with
physics-based models. *Hydrology and Earth System Sciences* **26**, 1063–1087 (2022).
- [53] Zhu, S. & Piotrowski, A. P. River/stream water temperature forecasting using artificial intelligence models:
a systematic review. *Acta Geophysica* **68**, 1433–1442 (2020).
- [54] Acuña Espinoza, E. *et al.* Analyzing the generalization capabilities of a hybrid hydrological model for
extrapolation to extreme events. *Hydrology and Earth System Sciences* **29**, 1277–1294 (2025). Publisher:
Copernicus GmbH.
- [55] Holmberg, E., Messori, G., Caballero, R. & Faranda, D. The link between European warm-temperature
extremes and atmospheric persistence. *Earth System Dynamics* **14**, 737–765 (2023).
- [56] Brunner, M. I., Götte, J., Schlemper, C. & Van Loon, A. F. Hydrological Drought Generation Processes
and Severity Are Changing in the Alps. *Geophysical Research Letters* **50**, e2022GL101776 (2023). [_eprint:
https://agupubs.onlinelibrary.wiley.com/doi/pdf/10.1029/2022GL101776](https://agupubs.onlinelibrary.wiley.com/doi/pdf/10.1029/2022GL101776).
- [57] Forzieri, G. *et al.* Ensemble projections of future streamflow droughts in Europe. *Hydrology and Earth
System Sciences* **18**, 85–108 (2014). Publisher: Copernicus GmbH.
- [58] Brunner, M., Farinotti, D., Zekollari, H., Huss, M. & Zappa, M. Future shifts in extreme flow regimes in
Alpine regions. *Hydrology and Earth System Sciences* **23**, 4471–4489 (2019).
- [59] Beniston, M. & Goyette, S. Changes in variability and persistence of climate in Switzerland: Exploring 20th
century observations and 21st century simulations. *Global and Planetary Change* **57**, 1–15 (2007).
- [60] Hoffmann, P., Lehmann, J., Fallah, B. & Hattermann, F. F. Atmosphere similarity patterns in boreal summer
show an increase of persistent weather conditions connected to hydro-climatic risks. *Scientific Reports* **11**,
22893 (2021).
- [61] Russo, S., Sillmann, J. & Fischer, E. M. Top ten European heatwaves since 1950 and their occurrence in the
coming decades. *Environmental Research Letters* **10**, 124003 (2015). Publisher: IOP Publishing.
- [62] Russo, S. *et al.* Magnitude of extreme heat waves in present climate and their projection in a
warming world. *Journal of Geophysical Research: Atmospheres* **119**, 12,500–12,512 (2014). [_eprint:
https://onlinelibrary.wiley.com/doi/pdf/10.1002/2014JD022098](https://onlinelibrary.wiley.com/doi/pdf/10.1002/2014JD022098).
- [63] Lhotka, O., Kyselý, J. & Farda, A. Climate change scenarios of heat waves in Central Europe and their
uncertainties. *Theoretical and Applied Climatology* **131**, 1043–1054 (2018).
- [64] Manning, C. *et al.* Increased probability of compound long-duration dry and hot events in Europe during
summer (1950–2013). *Environmental Research Letters* **14**, 094006 (2019). Publisher: IOP Publishing.
- [65] Geirinhas, J. L. *et al.* Recent increasing frequency of compound summer drought and heatwaves in Southeast
Brazil. *Environmental Research Letters* **16**, 034036 (2021). Publisher: IOP Publishing.
- [66] Råman Vinnå, L., Bigler, V., Schilling, O. S. & Epting, J. Multi-fidelity model assessment of climate change
impacts on river water temperatures, thermal extremes and potential effects on cold water fish in Switzerland.
*EGUsphere* 1–44 (2025).
- [67] Beniston, M. Warm winter spells in the Swiss Alps: Strong heat waves in a cold season? A study focusing
on climate observations at the Saentis high mountain site. *Geophysical Research Letters* **32** (2005).
- [68] Klingler, C., Schulz, K. & Herrnegger, M. LamaH-CE: LARge-SaMple DAta for Hydrology and Environmental
Sciences for Central Europe. *Earth System Science Data* **13**, 4529–4565 (2021).
- [69] Höge, M. *et al.* CAMELS-CH: hydro-meteorological time series and landscape attributes for 331 catchments
in hydrologic Switzerland. *Earth System Science Data* **15**, 5755–5784 (2023).
- [70] Schimanke, S. *et al.* CERRA sub-daily regional reanalysis data for Europe on single levels from 1984 to
present. (2021).
- [71] Sutanudjaja, E. H. *et al.* PCR-GLOBWB 2: a 5arcmin global hydrological and water resources model.
*Geoscientific Model Development* **11**, 2429–2453 (2018).
- [72] Hoch, J. M., Sutanudjaja, E. H., Wanders, N., van Beek, R. L. P. H. & Bierkens, M. F. P. Hyper-resolution
PCR-GLOBWB: opportunities and challenges from refining model spatial resolution to 1 km over
the European continent. *Hydrology and Earth System Sciences* **27**, 1383–1401 (2023).
- [73] Janzing, J. *et al.* Hyper-resolution large-scale hydrological modelling benefits from improved process repre-
sentation in mountain regions. *EGUsphere* 1–46 (2024). [accepted for publication].
- [74] van Jaarsveld, B. *et al.* A first attempt to model global hydrology at hyper-resolution. *Earth System*
*Dynamics* **16**, 29–54 (2025). Publisher: Copernicus GmbH.
- [75] Karger, D. N. *et al.* Climatologies at high resolution for the earth’s land surface areas. *Scientific Data* **4**,
170122 (2017). Publisher: Nature Publishing Group.
- [76] Karger, D. N. *et al.* Climatologies at high resolution for the earth’s land surface areas (CHELSA) V2.1
(2021). Repository: EnviDat.
- [77] Hobday, A. J. *et al.* A hierarchical approach to defining marine heatwaves. *Progress in Oceanography* **141**,
227–238 (2016).
- [78] Woolway, R. I. *et al.* Lake heatwaves under climate change. *Nature* **589**, 402–407 (2021).
- [79] Jensen, L. F. *et al.* Local adaptation in brown trout early life-history traits: implications for climate change
adaptability. *Proceedings of the Royal Society B: Biological Sciences* **275**, 2859–2868 (2008).
- [80] Butzge, A. J. *et al.* Early warming stress on rainbow trout juveniles impairs male reproduction but contrast-
ingly elicits intergenerational thermotolerance. *Scientific Reports* **11**, 17053 (2021).
- [81] Islam, M. J., Kunzmann, A. & Slater, M. J. Responses of aquaculture fish to climate change-induced extreme
temperatures: A review. *Journal of the World Aquaculture Society* **53**, 314–366 (2022).
**Acknowledgements**
The authors thank the Swiss Federal Office for the Environment and the Austrian Hydrographic Service for data
provision.
**Author contributions**
Note: *We make use of the double anonymized peer review. The author contributions are known by the editor.*
**Competing interests**
The authors declare no competing interests.
**Supplementary information**
Supplementary information is provided as a single separate file in PDF format.

Review of the manuscript:

This manuscript uses high-frequency observational datasets to investigate the response of riverine heatwaves to atmospheric heatwaves under different hydro-climatic conditions. The authors report that only 47% of atmospheric heatwaves lead to riverine heatwaves, and that the occurrence of riverine heatwaves often requires not only atmospheric forcing but also additional hydro-climatic conditions such as anomalous discharge and meltwater. Furthermore, the study presents a detailed case analysis of a river network to explore the interactions among riverine heatwaves, atmospheric heatwaves, and hydro-climatic variables.

Compared with previous studies, which primarily focused on long-term changes in riverine heatwave characteristics such as intensity, frequency and duration, then tended to attribute them in a simplified manner to atmospheric heatwaves, rising air temperatures, or discharge anomalies, this study provides a more systematic perspective. Specifically, it analyzes the hydro-climatic conditions under which atmospheric heatwaves are likely to trigger riverine heatwaves, thereby offering valuable insights into the driving mechanisms of riverine heatwaves.

While the manuscript addresses an important and timely topic, several aspects require clarification and further development:

1. The central question for me is how the authors define that riverine heatwaves are triggered by atmospheric heatwaves. Is this based solely on

temporal overlap? If so, I am not convinced this is a robust approach, this only reflects coincidence or probability rather than causality. Recent studies (Wang et al., 2024a, b; Sun et al., 2025) have used model simulations and controlled experiments to explicitly determine the role of atmospheric heatwaves in driving freshwater heatwaves. The authors should clarify their criteria and consider whether this approach is robust, because it is the base and core of this manuscript.

2. Line 87-88: The manuscript reports that the link between air and water temperatures is strongest in catchments at low elevations with a small snow fraction. Although Figure 1c seems to suggest this pattern, the figure needs added relationship value and statistical significance to support the conclusion.

3. Line 110: The suitability of the term “decoupling” is questionable. Previous studies (e.g., Cassie., 2006) have shown that air and water temperatures follow an S-shaped relationship: linear in the mid-range but nonlinear at low and high extremes, where river water temperature ceases to increase proportionally with air temperature. Since riverine heatwaves are extreme events, it is unlikely that these processes represent a true “decoupling” between atmospheric and riverine heatwaves. The authors should reconsider their terminology and provide a more accurate interpretation.

4. Line 127: The manuscript emphasizes cases where atmospheric

heatwaves coincide with riverine heatwaves. However, it would also be valuable to examine cases where riverine heatwaves occur in the absence of atmospheric heatwaves. Do such cases exist? If so, comparing these with the co-occurrence cases could provide important insights into additional hydro-meteorological influences.

5. Line 154: The described phenomenon is puzzling. In general, solar radiation is an important driver of river water temperature, while high humidity, which as authors described is often associated with rainfall, reduces solar radiation reaching the water surface and thereby suppresses warming. In the discussion, the authors suggest that high humidity and reduced radiation accompany warmer temperatures, rainy, or stormy weather, which may lead to riverine heatwaves. However, it is unclear whether rainy or stormy conditions trigger or suppress riverine heatwaves, and I am not aware of references supporting this interpretation. A more detailed analysis, supported by relevant literature, is needed to provide a convincing explanation of this mechanism.

6. Line 159: Why were these five stations in Drau River chosen? Were they the only locations where this phenomenon was observed, or were there other criteria guiding the selection? A more transparent justification would be helpful.

Overall, while the study provides valuable insights, these clarifications are necessary before the manuscript can be considered for

publication.

Reference:

Wang, W. et al. The impact of extreme heat on lake warming in China. *Nat Commun* **15**, 70 (2024).

Wang, X. et al. Disproportionate impact of atmospheric heat events on lake surface water temperature increases. *Nat. Clim. Chang.* (2024)
doi:10.1038/s41558-024-02122-y.

Sun, J. et al. Impact of extreme atmospheric heat events on river thermal dynamics and heatwaves. *Journal of Hydrology* **659**, 133292 (2025).

Caissie, D., 2006. The thermal regime of rivers: a review. *Freshw. Biol.* 51 (8), 1389-1406.

Review of the manuscript:

The authors have carefully responded to the comments raised in the first round of review, and the revisions and responses are generally clear and effective in addressing my previous concerns. The overall quality of the manuscript has improved substantially, and the revised version is clearly stronger than the original submission. I also appreciate the authors' decision to rely on in situ observations, which represents a clear strength of this study. Given the limited number of studies that investigate riverine heatwaves based on observed river temperature data, this observational perspective provides an important and valuable contribution to the field.

The methodology adopted in the manuscript primarily identifies potential "triggering" mechanisms based on the temporal overlap between atmospheric and riverine heatwaves. This framework appears to focus more on describing compound or co-occurring heatwave events rather than establishing a clear causal relationship. In addition, the thermal response of river systems often exhibits pronounced lag effects due to hydrological and geomorphological controls, which may not be fully captured by simple temporal concurrence analyses. This point was also raised in my first-round review. At the same time, even if this approach may currently represent the best feasible methodology based on available observational datasets, its robustness and inherent limitations still warrant careful discussion. While the authors argue that this observation-based approach is preferable to

previous modeling studies. I also acknowledged that numerical models, despite introducing their own sources of uncertainty, remain the most direct method for isolating atmospheric forcing through controlled, variable-specific experiments.

Therefore, I strongly recommend that the authors include a dedicated paragraph or section to introduce the limitations of the current methodology, as well as a balanced comparison of the strengths and weaknesses of this study approach relative to modeling-based methods used in previous studies.